# Neoantigen-specific CD8 T cell responses in the peripheral blood following PD-L1 blockade might predict therapy outcome in metastatic urothelial carcinoma

Jeppe Sejerø Holm[1,9], Samuel A. Funt [2,3,9], Annie Borch[1], Kamilla Kjærgaard Munk[1], Anne-Mette Bjerregaard [1], James L. Reading [4], Colleen Maher [2,3,5], Ashley Regazzi [2,3], Phillip Wong [2,3,5], Hikmat Al-Ahmadie [6], Gopa Iyer [2,3], Tripti Tamhane [1], Amalie Kai Bentzen[1], Nana Overgaard Herschend[1], Susan De Wolf[2], Alexandra Snyder[2,3], Taha Merghoub [2,3,5], Jedd D. Wolchok [2,3,5,7], Morten Nielsen [8], Jonathan E. Rosenberg[2,3], Dean F. Bajorin [2,3,9] & Sine Reker Hadrup[1,9 ✉]

CD8$^+$ T cell reactivity towards tumor mutation-derived neoantigens is widely believed to facilitate the antitumor immunity induced by immune checkpoint blockade (ICB). Here we show that broadening in the number of neoantigen-reactive CD8$^+$ T cell (NART) populations between pre-treatment to 3-weeks post-treatment distinguishes patients with controlled disease compared to patients with progressive disease in metastatic urothelial carcinoma (mUC) treated with PD-L1-blockade. The longitudinal analysis of peripheral CD8$^+$ T cell recognition of patient-specific neopeptide libraries consisting of DNA barcode-labelled pMHC multimers in a cohort of 24 patients from the clinical trial NCT02108652 also shows that peripheral NARTs derived from patients with disease control are characterised by a PD1$^+$ Ki67$^+$ effector phenotype and increased CD39 levels compared to bystander bulk- and virus-antigen reactive CD8$^+$ T cells. The study provides insights into NART characteristics following ICB and suggests that early-stage NART expansion and activation are associated with response to ICB in patients with mUC.

[1] Experimental and Translational Immunology, Health Technology, Technical University of Denmark, Kgs. Lyngby, Denmark. [2] Department of Medicine, Memorial Sloan Kettering Cancer Center, New York, NY 10065, USA. [3] Weill Cornell Medical College, New York, NY 10065, USA. [4] Cancer Immunology Unit, Research Department of Hematology and Cancer Research UK, Lung Cancer Centre of Excellence, University College London Cancer Institute, London, UK. [5] Parker Institute for Cancer Immunotherapy, San Francisco, CA, USA. [6] Department of Pathology, Memorial Sloan Kettering Cancer Center, New York, NY, USA. [7] Human Oncology and Pathogenesis Program, Memorial Sloan Kettering Cancer Center, New York, NY 10065, USA. [8] Section of Bioinformatics, Health Technology, Technical University of Denmark, Kgs. Lyngby, Denmark. [9] These authors contributed equally: Jeppe Sejerø Holm, Samuel A. Funt, Dean F. Bajorin, Sine Reker Hadrup. ✉email: sirha@dtu.dk

The anti-tumor T cell response induced by ICB of the programmed death 1 (PD-1) / programmed death-ligand 1 (PD-L1) axis can result in deep and durable responses in patients with a variety of metastatic cancers[1–5]. In pre-treatment tumors, a high mutational burden is correlated with a beneficial response to immune checkpoint blockade (ICB) across multiple indications, which is thought to be, at least in part, due to increased presentation of exogenous neoantigens displayed by major histocompatibility complex (MHC) class I to CD8+ T cells reinvigorated by ICB[6–9]. Indeed, a growing body of evidence suggests that CD8+ T cells not only infiltrate responding tumors but also undergo rapid and robust proliferation in the peripheral blood of patients following treatment with ICB[10–17]. With regard to the specificity of this CD8+ T cell response, the interrogation of neoantigen recognizing T cells (NARTs) has mostly been limited to tumor tissue[18]. Importantly, since the relevant neoantigens are unique for the individual patient, detailed NART analysis requires the prediction of potential neoantigens and T cell screening with a unique set of neoantigens for each patient. This effort has been hindered by technological barriers limiting the analytic range required to comprehensively characterize the vast number of neoantigens potentially presented as well as the diversity of potential human leukocyte antigen class I (HLA) genotypes.

To address these issues, we focused on a previously reported cohort of patients who were treated with the anti-PD-L1 antibody atezolizumab for metastatic urothelial carcinoma (mUC)[19]. Clinical outcome of mUC has significantly benefited from the introduction of ICB treatment, with approximately 20% of patients with previously fatal disease experiencing long-term survival[20,21]. Additionally, long-term follow-up was obtained for the patients under study ($n = 24$), with some remaining progression free 5+ years after treatment initiation. Peripheral blood samples collected pre-, during-, and post-treatment were comprehensively screened using patient-specific neopeptide-MHC (pMHC) multimer libraries, labeled with DNA barcodes[22], allowing for high-throughput detection of CD8+ T cell populations recognizing any such neopeptides in one parallel reaction. We interrogated NART dynamics and phenotype using this novel technique and evaluated for associations with clinical outcome.

## Results

**Neoepitope prediction and T cell screening.** Neoepitopes were predicted for each individual patient from Whole Exome Sequencing (WES) and RNAseq by use of the MuPeXI platform[23]. Potential neopeptide candidates were selected based on their respective MHC-I binding affinity and expression level, and the experimental availability of the recombinant MHC-I molecules relevant for the given peptide. From the 24 patients, 56 different HLA ABC-haplotypes are represented, and at the time of analysis 31 of these were available for pMHC multimer generation. On average, four HLA haplotypes were covered for each patient. All HLA-feasible neopeptides with a predicted Eluted Ligand (EL)%Rank score[24] <0.5 and expression level >0.1 transcripts per million (TPM) were selected, yielding between 14-587 HLA-binding neopeptides per patient (Fig. 1a). To allow complete evaluation of potential neoepitopes in patients with low mutational burden, additional neopeptides with higher EL%Rank score, derived from genes with expression level >0.1 TPM were included, based on lowest EL%Rank score, until a minimum of 200 neopeptides was reached for T cell recognition analyses per patient. As a result, each patient was analyzed using pMHC multimer libraries displaying between 200-587 patient-unique neopeptides (Fig. 1b). In total, 6237 HLA-feasible neopeptides

across the 24 patients were predicted and included for T cell analyses (Table 1).

In this patient cohort, neither the tumor mutational burden (TMB) nor number of predicted neopeptides with EL%Rank <0.5 was predictive of ICB outcomes (Fig. 1c, d). Hence, we evaluated for the presence of circulating NARTs to gain greater insights into the nature of anti-tumor immunity following initiation of anti-PD-L1 therapy.

**Presence of NARTs in patient PBMC samples.** Based on the selected neopeptides, barcoded pMHC multimer libraries were generated for each patient matching their HLA-type (Fig. 2a). In addition to neopeptides, one to 17 HLA-matching virus-derived peptides from cytomegalovirus (CMV), Epstein-Barr viral (EBV), and influenza (FLU, together; CEF) were included in each patient's peptide library for internal assay validation and to compare NARTs to virus-antigen reactive T cells (VARTs; Table 1). T cell recognition was examined by peripheral blood mononuclear cell (PBMC) staining and sorting of a pMHC associated barcode in the multimer-binding T cell population. Significant T cell recognition of a given neoepitope (NART response) was defined as a $Log_2$ fold change ($Log_2FC$) >2 and false-discovery rate (FDR) < 0.1, based on previous investigations[22].

PBMC samples, collected just prior to administration of atezolizumab, were screened for the presence of NARTs ($n = 85$ samples, median = 3 samples per patient) at the indicated time points (Table 1 + Supplementary Table 1). Of note, PBMCs were available from sampling up to >231 weeks after treatment initiation for six long-term responders.

Representative outputs from patient screenings for NARTs is depicted in Fig. 2b and Supplementary Fig. 2a, b. For patient #2389 (ongoing complete response [CR] per best response evaluation criteria in solid tumors [RECIST] version 1.1 criteria, >280 weeks after treatment initiation), the T cell recognition ($Log_2FC$) is depicted for each of the pMHC multimers evaluated in this patient ($n = 203$; Fig. 2b). The outcome is listed according to the time points of blood sampling and grouped by the evaluated HLA molecules. An emergence of new T cell populations recognizing neoepitopes presented on HLA-A*01:01 and HLA-B*40:01 is seen from pre-treatment to 3 weeks post-treatment, while a T cell response towards a FLU epitope presented on HLA-A*01:01 (VSDGGPNLY) is detected at the majority of screened timepoints. Based on additional sample availability, 65 NART responses detected at 3- or 9 weeks post-treatment were interrogated using neo-pMHC tetramers (Fig. 2c). 50 NART responses were validated, whereas an additional nine responses were borderline detectable (Supplementary Fig. 3). These borderline detectable responses may represent low-affinity NARTs, likely detected only by using DNA barcode-labelled multimers due to enhanced sensitivity compared to conventional tetramer-based detection for detection of such T cells[22]. In parallel with the screening of patient samples, a selected set of healthy donor (HD) PBMC samples were evaluated to validate all pMHC multimer libraries based on the included CEF-peptides (Supplementary Fig. 2c). A few pMHC multimer complexes demonstrated unspecific binding in all samples and were excluded from further analyses.

At the pre-treatment time point, T cells recognizing neoepitopes were detected in the 18 of 24 of patients (median 2; range 0 to 10 NART responses per patient). After 3 weeks of atezolizumab treatment, T cells recognizing neoepitopes were detected in the 17 of 22 patients (median 3; range 0 to 13 NART responses per patient; Table 1). 18 of 22 patients, including four with no detectable pre-treatment NARTs, developed NARTs post-treatment that were not originally present pre-treatment. 45 of

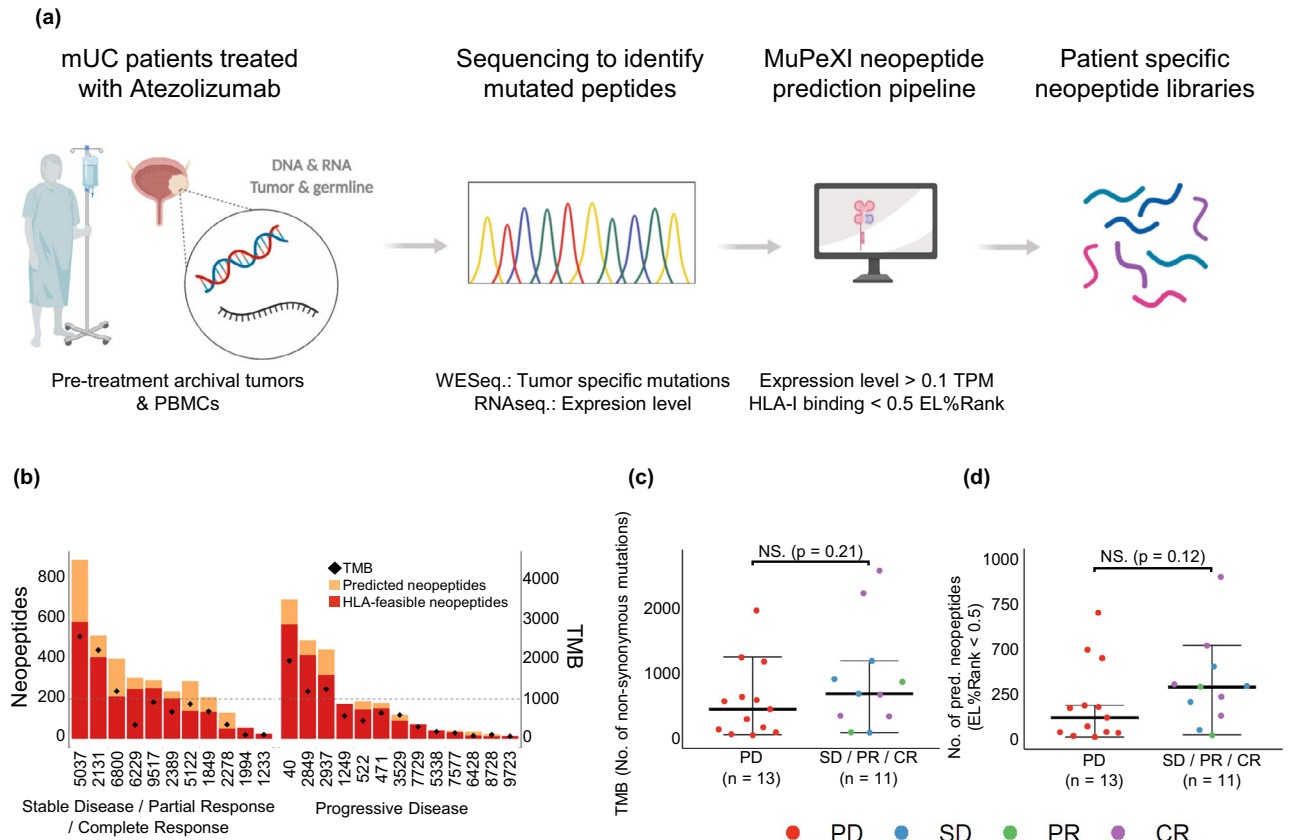

**Fig. 1 Prediction of patient-specific neopeptides. a** Overview of workflow for identification and prediction of mutation-derived neopeptides in mUC patients. Created with BioRender.com. **b** Number of predicted and HLA-feasible neopeptides with EL%Rank <0.5, and TMB (diamond) in the patient cohort. Patients grouped according to best RECIST 1.1 criteria, PD ($n = 13$ patients; median = 123, range = 16–700 neopeptides with EL%Rank <0.5 predicted) and SD/PR/CR patients ($n = 11$ patients; median = 292, range = 26–898 neopeptides with EL%Rank <0.5 predicted). Dotted line represents minimum 200 neopeptides included in panel for each patient. **c**, **d** TMB and number of prediced neopeptides with EL%Rank <0.5 for patients with PD compared to SD/PR/CR patients. For (**c**, **d**) groups were compared using non-parametric two-sided Mann–Whitney test and data is presented as median values ± largest/smallest value within upper/lower quartile ± 1.5 IQR. NS. Not Significant. Source data are provided as a Source Data file.

unique 148 NART responses were observed at multiple time-points (Supplementary Table 1). There was no immediate association between the number of unique NART responses detected throughout treatment and the number of predicted and evaluated neoepitopes (Fig. 2d). This indicates that other parameters, beyond TMB, influences the capacity to drive such T cell responses upon ICB initiation.

**Enhancement of NART responses three weeks post-treatment is associated with improved clinical outcome**. At the pre-treatment time point, no association between the number of NART responses and best RECIST response was observed. However, a difference in the kinetics of NART responses was noted over the course of treatment, with NART populations increasing three weeks post-treatment and then contracting in the majority of patients with disease control (defined as patients with SD, PR, and CR) but not in patients with progressive disease (PD; Fig. 3a). Indeed, at the three week post-treatment time point, patients with disease control tended to have a higher number of NART responses compared to patients with PD (Fig. 3a, $p = 0.067$), and also when comparing patients with a CR versus PD (Fig. 3b, c). No significant differences between patient response groups were observed at subsequent time points. The change (*delta*) in number of detected NART responses between pre-treatment and three weeks post-treatment was calculated to better approximate patient-specific NART dynamics (Fig. 3d–e). A significant increase in ΔNART responses was observed for

patients with disease control compared to those with PD (Fig. 3d, $p = 0.012$), with CR patients experiencing the largest ΔNART responses compared to PD patients (Fig. 3e, $p = 0.022$). Although a substantial smaller library of CEF-derived epitopes was included in the analyses compared to neoepitopes, no changes in the VART response repertoire were observed during treatment (Supplementary Fig. 5). The frequency of NART responses was estimated based on pMHC multimer staining and the fraction of barcode reads assigned to the given populations (see Materials and Methods). The sum of estimated frequencies (SEFs) across all patient samples ranged between 0.01% and 3.9% ($n = 62$, 0.55% average) (Supplementary Fig. 4a). Individual response estimated frequencies range from 0.01% up to ~2.83% ($n = 221$), with >98% of response frequencies being below 1% (mean = 0.15 %, median = 0.048%). Hence, SEFs in patient samples are not skewed by single, large NART response frequencies. Neither the absolute SEF nor the change in SEF from baseline to 3-weeks post treatment was associated to clinical outcome (Fig. 3f–i). Thus, based on this evaluation, a substantial difference in the number of NART responses was observed between patients with PD and those with CR, and the breadth of the NART repertoire rather than the combined estimated frequencies of such populations was the key parameter associated to favorable clinical outcome in the setting of ICB treatment. The increase *delta* in number of NART responses from pre- to 3 week post-treatment for patients with disease control indicate that these patients tended to rapidly raise a broader T cell neoantigen recognition repertoire post-treatment.

**Table 1 Patient clinical data, feasible HLA-I information, neo- and viral peptides library size and no. of T cell responses for all patients. No. of T cell responses indicated for each patient at respective timepoints.**

| Patient | Clinical data | Feasible HLA-I | | | HLA-I-feasible neopeptides | | Viral peptides | T cell neoepitope responses, per time point | | | | | | |
| | Outcome (Best RECIST 1.1) | HLA-A | HLA-B | HLA-C | Predicted (EL%Rank <0.5) | Library size | Library size | Pre-treatment | 3 weeks | 9 weeks | 20-29 weeks | 49-73 weeks | 156-166 weeks | 231+ weeks |
|---|---|---|---|---|---|---|---|---|---|---|---|---|---|---|
| #40 | PD[2] | A0201/A3201 | B1801 | C0401/C0701 | 576 | 576 | 9 | 5 | – | – | – | – | – | – |
| #9723 | PD[2] | A0101/A2402 | B0702/B0801 | C0701/C0702 | 16 | 200 | 16 | 1 | – | – | – | – | – | – |
| #1249 | PD[2] | A0301/A6801 | B3501/B4402 | C0401/C0501 | 177 | 200 | 12 | 1 | 2 | – | – | – | – | – |
| #2937 | PD | A3201 | B1801/B3503 | C0401 | 322 | 322 | 1 | 9 | 5 | 3 | – | – | – | – |
| #1994 | SD | A0301 | B0702 | C0702 | 56 | 200 | 9 | 3 | 2 | 3 | – | – | – | – |
| #5221 | PD | A2402/A3001 | B5701 | C0602 | 150 | 200 | 6 | 2 | 3 | 2 | – | – | – | – |
| #6428 | PD | A2601 | B2705 | C0202 | 18 | 200 | 1 | 0 | 0 | 0 | – | – | – | – |
| #7577 | PD | A0101/A0201 | B5101 | C0602/C0701 | 35 | 200 | 14 | 1 | 2 | 0 | – | – | – | – |
| #8728 | PD | – | B4403/B5101 | | 14 | 200 | 4 | 5 | 2 | 0 | – | – | – | – |
| #471 | PD | A0201/A6801 | B3503/B5101 | C0401 | 155 | 200 | 11 | 0 | 3 | 7 | – | – | – | – |
| #3529 | PD | A0205/A2402 | | C0602 | 92 | 200 | 5 | 5 | 0 | 0 | – | – | – | – |
| #5338 | PD | A0201/A2402 | B4001/B4002 | C0202/C0304 | 44 | 420 | 16 | 1 | 1 | 2 | – | – | – | – |
| #2849 | PD | A0201/A0301 | B0702/B2705 | C0702 | 420 | 420 | 17 | 5 | 3 | 5 | – | – | – | – |
| #7729 | PD | A0101/A0301 | B0801/B1302 | C0602/C0701 | 75 | 200 | 15 | 2 | 7 | 3 | 3 | – | – | – |
| #9517 | SD | A0101/A0301 | B3501/B3801 | C0401 | 254 | 254 | 13 | 5 | 9 | 5 | 4 | – | – | – |
| #1849 | SD | A0101/A1101 | | C0401/C0602 | 136 | 200 | 7 | 10 | 5 | 14 | – | – | – | – |
| #6800 | SD[1] | A0301 | B1501 | C0401 | 214 | 214 | 10 | 0 | 13 | 1 | – | – | – | – |
| #6229 | CR | A3001/A3201 | B1302 | C0401/C0602 | 250 | 250 | 4 | 5 | 6 | 3 | – | 2 | – | – |
| #5037 | CR | A2601 | B3801/B4002 | C0202 | 587 | 587 | 2 | 2 | 4 | 4 | – | 2 | 0 | 0 |
| #2389 | CR | A0101/A2402 | B3801/B4001 | C0304 | 203 | 203 | 9 | 2 | 4 | 3 | – | 0 | 0 | 1 |
| #2131 | CR | A0201 | B0801 | C0202/C0701 | 411 | 411 | 14 | 0 | 2 | 1 | – | 6 | 5 | 4 |
| #2278 | CR | A0201/A2402 | – | C0202 | 52 | 200 | 13 | 1 | 2 | – | – | 3 | 1 | 0 |
| #1233 | PR | A1101/A2601 | B2705/B3501 | C0202/C0702 | 26 | 200 | 8 | 0 | 0 | 2 | – | 2 | 1 | 0 |
| #5122 | PR | A2402/A3201 | B3801 | – | 140 | 200 | 4 | 1 | 0 | 0 | – | 0 | 0 | 0 |

A dash (–) indicates no sample screening at the given time point. Clinical outcome determined from Best RECIST 1.1 criteria during therapy. [1]TCRb analysis for tumor not performed due to failed sequencing quality control. [2]Patient only scanned at baseline. Source data are provided as a Source Data file.

As described, predicted neoepitopes were originally included based on an EL%Rank score <0.5 selection criteria, but for 15 patients the neoepitope library sizes were extended to reach a minimum of 200 neoepitopes. We therefore evaluated if the potential differences in neoepitope characteristics and library size would influence our findings. Comparable analyses were conducted including only neoepitopes with EL%Rank score <0.5 and expression level >2 TPM or all library sizes were set to 200. For both such analyses, we observed similar trends as given for the total neopeptide libraries (Supplementary Fig. 4b±d). Hence, no bias in our findings was introduced by the original selection criteria.

Separating patients based according to Durable Clinical Benefit (DCB; progression-free survival, PFS >6 months) did not yield a significant difference in NART response numbers (Supplementary Fig. 4e). Furthermore, although not statistically significant, a trend for improved PFS and overall survival (OS) was seen for the patients with higher ΔNART response numbers from pre- to 3 weeks post-treatment (>median 0 ΔNART responses; Supplementary Fig. 4f–h).

**Peripheral blood TCR metrics display similar kinetics as NARTs.** TCR diversity and clonality for PBMCs and TILs have been previously shown to be correlated with response to ICB[14,25,26]. For this cohort, a higher fraction of the T cell clones present in tumor were seen to expand in the blood 3 weeks post-treatment for patients with DCB[19]. We observed a rapid spike in bulk TCR clonality early post-treatment, similar to the observed development in the number of NART responses (Supplementary Fig. 4i), even though changes in bulk TCR clonality or diversity did not differentiate patients with and without response to therapy (data not shown). Although bulk TCR sequencing does not identify the antigen specificity of individual clones, the parallel kinetics between the NART response development and TCR clonality for patients with a favorable clinical outcome is noteworthy. This further supports the contention that clonal expansion and T cell reinvigoration occurs early post-treatment following ICB.

**Phenotypic characterization of NARTs indicate increased proliferation of NARTs in patients with disease control.** To characterize the phenotypic profile of NARTs, a custom multi-color flow cytometry antibody panel was designed to characterize T cell differentiation, exhaustion, activation, and migration (Supplementary Fig. 6a). Both phycoerythrin (PE)-neoepitope multimers and viral pMHC multimers conjugated to allophyco-cyanin (APC) were included in the panel to further differentiate the phenotypic profiles of NARTs and VARTs. pMHC multimer-binding T cells were sorted and barcodes sequenced for epitope reactivity, whilst in parallel, the phenotypic profile of NARTs and VARTs were characterized. Patient PBMC samples from pre-treatment and at 3 weeks and 231+ weeks post-treatment were selected based on the initial early NART-response, and patient sample and multimer library availability (n = 34).

Data was visualized using Uniform Manifold Approximation and Projection (UMAP) dimensionality reduction plugin. Variations in population distribution were observed when faceting UMAPs by patient, either pre- to post-treatment or disease control versus PD patients (Fig. 4a, b). Populations that were enriched in density post-treatment were characterized by expression of Ki67, PD-1, and in part CD39 (Fig. 4c). In particular, Ki67 and PD-1-expressing NARTs appeared to be enriched post-treatment in disease control patients compared to PD patients. In contrast, NARTs expressing CD57 appeared more frequent in PD patients post-treatment. Hence, guided by the

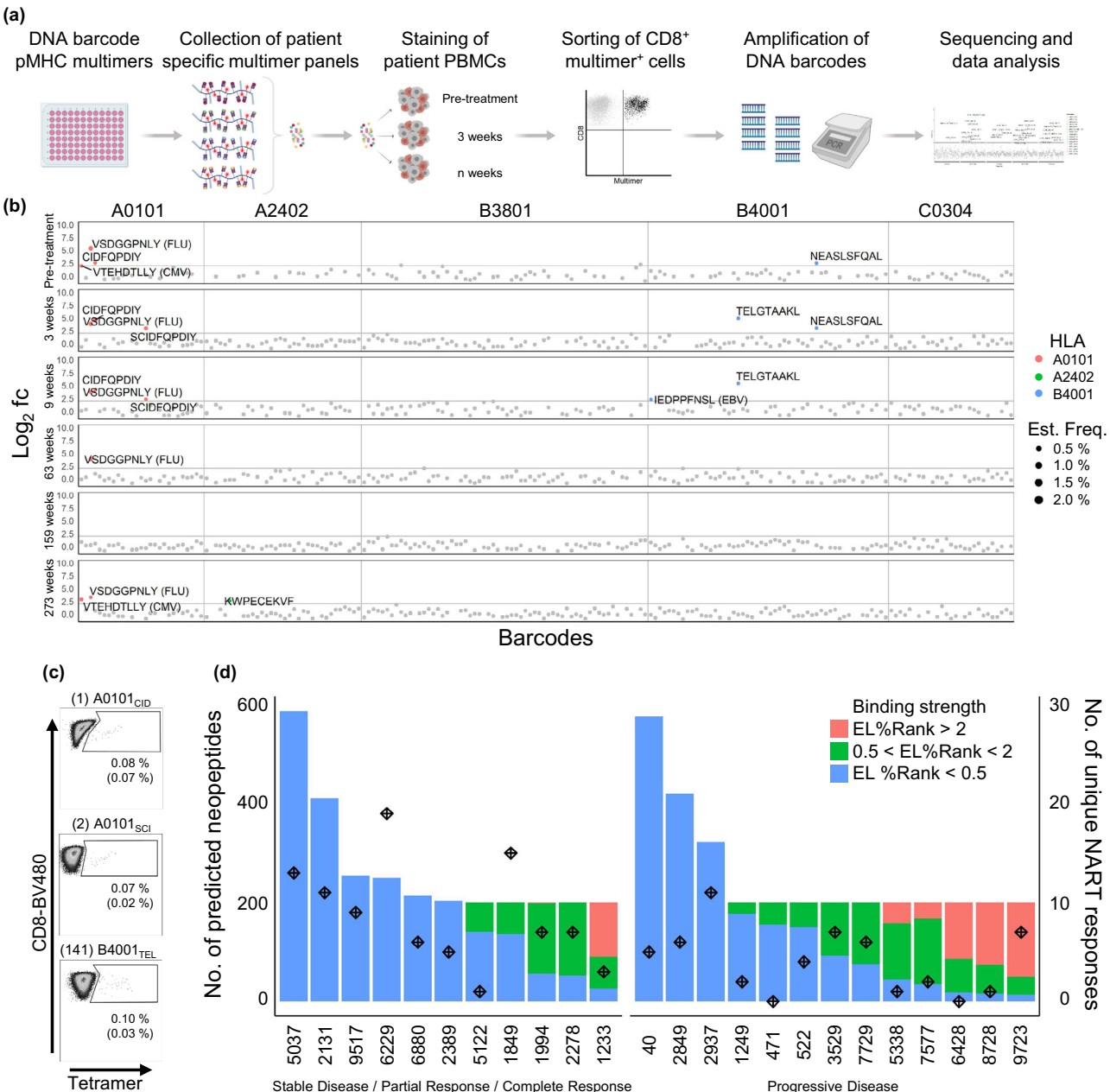

**Fig. 2 Detection of neoepitope-reactive T cell responses in mUC cancer patients. a** Overview of workflow for detection of neoantigen specific T cells in mUC cancer patients. Experimental steps include building of DNA-barcoded pMHC multimer libraries from predicted neopeptides, collection of multimer panels, staining of patient PBMCs with multimers, sorting of multimer-binding CD8+ T cells, amplification and sequencing of DNA barcodes, and data analysis. Created with BioRender.com. **b** Representative output from patient #2389, screening of detected NART responses. $Log_2$ fold change (fc) of sequenced pMHC associated barcodes enriched by T cell sorting over the input library at each timepoint: pre-treatment, three weeks, nine weeks, 63 weeks, 159 weeks and at 273 weeks post-treatment. Labelled points represent significantly enriched pMHC co-attached barcodes with $Log_2$ fc >2, count fraction >0.1% and $p < 0.001$, determined as T cell responses, among the sorted multimer positive CD8+ T cell populations. T cell responses are colored based on peptide-presenting HLA-type, text labelled with peptide sequence, and sized according the estimated frequency of the peptide-recognizing T cell population. If the peptide is derived from a virus, the virus is annotated in brackets, otherwise it is a mutation-derived neoepitope. Grey points represent non-enriched barcodes. Horizontal line at $Log_2$ fc = 2. Vertical line separating peptide-presenting HLA-types. **c** Tetramer validations of three NART responses in patient #2389 at 9-weeks post-treatment. CD8+ cells shown, gated for tetramer+ populations. Frequency of tetramer pMHC population given in percentage, together with estimated frequency derived from multimer screening given in brackets. **d** Library sizes of predicted HLA-I-feasible neopeptides per patient, colored bars according to predicted HLA-I binding strength, compared to total no. of unique NART responses across screened samples for each patient (stars). Source data are provided as a Source Data file.

signatures from the UMAP, the frequency of the parameters that appeared increased in disease control patients post-treatment, i.e. KI67, CD39 and PD-1, were quantified for 'bulk CD8 T cells', 'NARTs', and 'VARTs' in the individual patients and the evaluated time-points, based on the full dataset. We observed an increase in the frequency of Ki67+ (bulk CD8 $p = 0.00034$, NARTs $p = 0.0054$, VARTs $p = 0.001$) and PD-1+ (bulk CD8 $p = 0.041$, NARTs $p = 0.043$) CD8 T cells from baseline to three

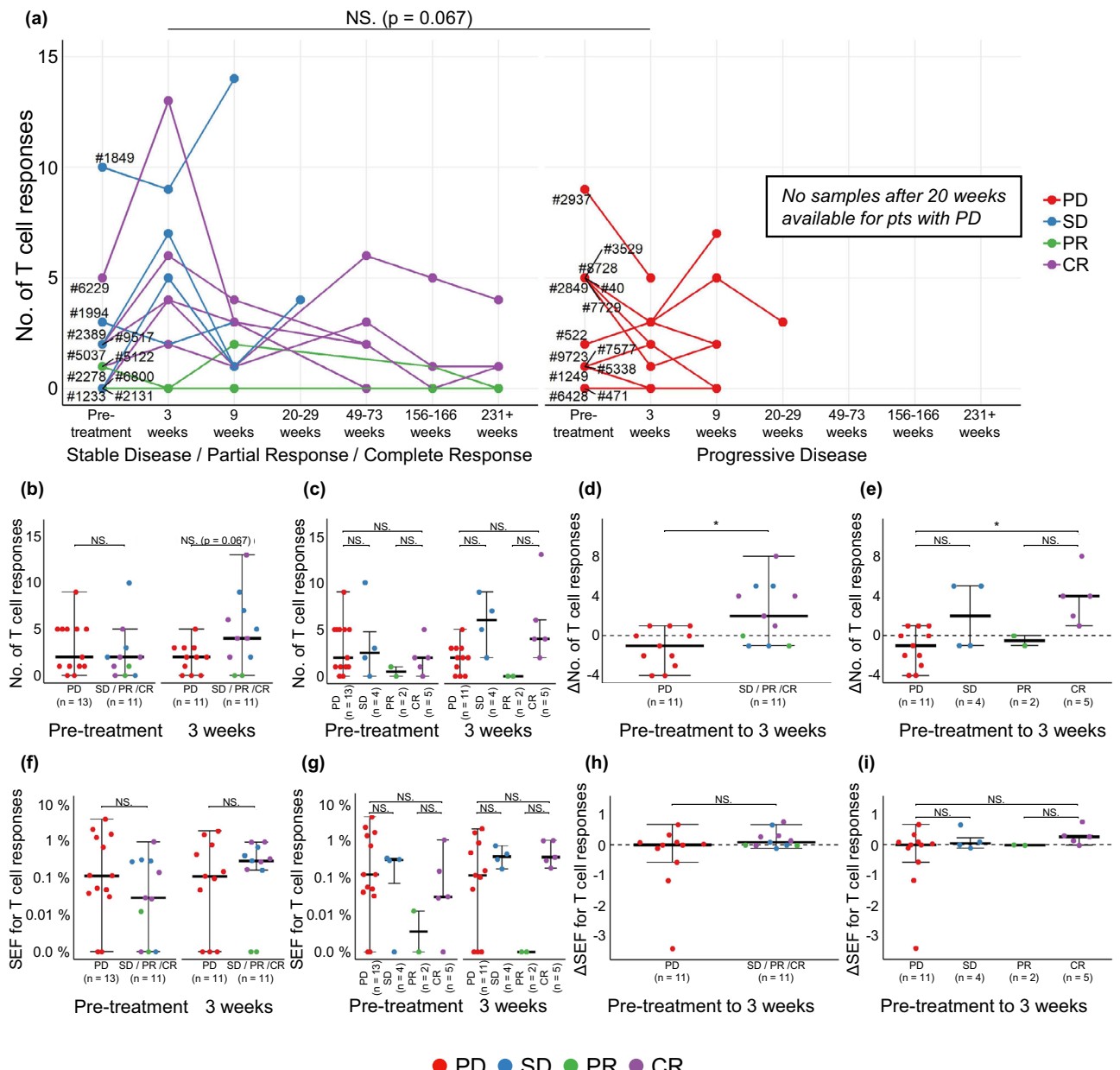

**Fig. 3 Impact of NARTs in ICB. a** No. of detected NART responses for all patients over course of therapy. Filled circles at respective time points indicate screening of the given sample. Patients are clustered according SD, PR, and CR ($n = 11$ patients) and with PD ($n = 13$ patients). **b–c** Number of detected NART responses prior to treatment ($n = 24$ patients) and at three weeks post-treatment ($n = 22$ patients) for each patient, **d–e** the change in number of NART responses between pre- and 3 weeks post-treatment for patients with minimum two time points sampled ($n = 22$ patients), **f–g** SEF for NART populations at pre- and at 3 weeks post-treatment, and (**h–i**) change in SEF for NART populations between pre- and three weeks post-treatment, in all figures evaluated as PD compared to SD/PR/CR patients or individual Best RECIST 1.1 groups. For (**b**), (**d**), (**f**) + (**h**) groups were compared using non-parametric two-sided Mann–Whitney test and Kruskal–Wallis Dunn's multiple comparison test for (**c**), (**e**), (**g**) + (**i**). For (**b–i**), data is presented as median values ± largest/smallest value within upper/lower quartile ± 1.5 IQR. NS. Not Significant, *$p < 0.05$. Source data are provided as a Source Data file.

weeks post-treatment across all evaluated subpopulations (Fig. 4d); indicating a general signature of T cell activation as a consequence of ICB. Importantly, this increase was almost exclusively observed for patients with disease control, with a significance for both Ki67+ bulk CD8 T cells ($p = 0.00088$), Ki67+ NARTs ($p = 0.011$), and Ki67+ VARTs ($p = 0.0091$). PD-1+ bulk CD8 T cells were increased slightly from pre- to post-treatment for PD patients ($p = 0.03$; Fig. 4e), but should be reflected based on a complete absence of PD-1+ bulk CD8 T cells prior to treatment initiation. In the VART population only, we

observed a marginal, non-significant increase in T cell activation by Ki67+ and PD-1+ in the PD group (Fig. 4e). It is evident that several patients in the disease control group have elevated levels of PD-1+ CD8 T cells, especially within 'bulk CD8' and 'NART', prior to therapy (bulk CD8 pre-treatment $p = 0.0089$; NARTs pre- $p = 0.02$; Fig. 4f). A similar enhanced level of both Ki67+ and PD-1+ 'bulk CD8' and CD39+, Ki67+ and PD-1+ NARTs is observed at 3wks post-treatment, although only significant for PD-1+ NARTs ($p = 0.028$; Fig. 4f). Interestingly, we observed that triple-positive Ki67+ PD-1+ CD39+ CD8 T cells (Bulk, NARTs

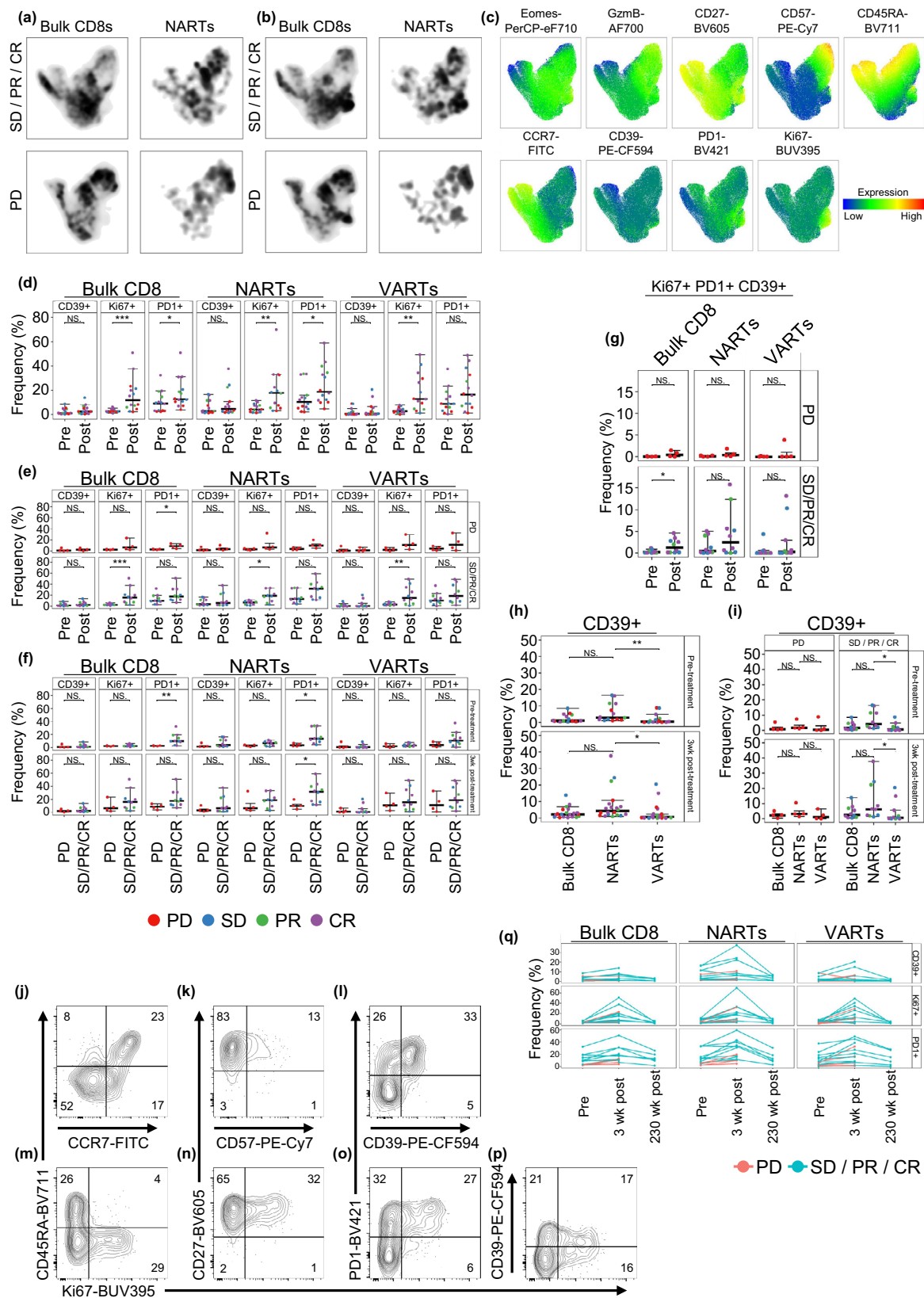

or VARTs) were completely absent in the PD group (Fig. 4g). Lastly, at 3 weeks post-treatment, up to 60% of NARTs are Ki67+CD45RA− cells and tend to constitute a larger subpopulation for patients with disease control (Supplementary Fig. 6b), with the majority of the cells being CD27+ rather than CD57+,

implying a state of activation rather than terminal differentiation (Supplementary Fig. 6c). No difference in frequencies of terminally differentiated, CD57+CD45RA+GzmB+ triple-positive cells were seen between patient groups or during treatment (Supplementary Fig. 6e, f). Detectable NARTs primarily comprise of T_Em cells, with

**Fig. 4 Phenotypic characterisation of bulk CD8 T cells, NARTs and VARTs. a–b** UMAPs of bulk concatenated CD8 + T cells and NARTs at (**a**) pre-treatment and (**b**) 3-week post-treatment for patients with SD/PR/CR ($n = 8$ patients) and PD ($n = 4$ patients). **c** Expression of phenotype markers on UMAP. Cells expressing similar parameters are clustered based on expression patterns. **d–f** Parent population frequencies of selected and (**g**) for triple-positive Ki67[+] PD-1[+] CD39[+] bulk CD8, NART and VART subpopulations for each patient, either at pre- ($n = 14$ patients) and 3 weeks post-treatment ($n = 14$ patients) and between patient with SD/PR/CR ($n = 10$ patients) and PD ($n = 4$ patients). **h–i** CD39[+] frequencies for bulk CD8 T cells, NART, and VART subpopulations at pre- and post-treatment for patients with SD/PR/CR ($n = 10$ patients) and PD ($n = 4$ patients). **j–p** Representative example of flow contour plots (5%, outliers shown) of key parameters for the NART subpopulation at three-week post-treatment for patient #2389 (DCB; Best RECIST 1.1 CR). **q** Ki67[+], CD39[+], and PD-1[+] subpopulation frequencies for bulk CD8 T cells, NARTs, and VARTs at pre-treatment ($n = 14$ patients) and at three weeks ($n = 14$ patients) and 230+ weeks post-treatment ($n = 6$ patients). For (**d–g**) groups were compared using non-parametric two-sided Mann–Whitney test and Kruskal–Wallis Dunn's multiple comparison test for (**h**) + (**i**). For (**d–i**), data is presented as median values ± largest/smallest value within upper/lower quartile ± 1.5 IQR. NS. Not Significant, *$p < 0.05$, **$p < 0.01$, ***$p < 0.001$. wk: week, Pre: Pre-treatment, Post: Post-treatment. In (**d**), (**e**) and (**g**) Post: three weeks post-treatment. Source data are provided as a Source Data file.

a smaller fraction of Naïve NARTs (Supplementary Fig. 6g). This indicates that T cell recognition based on NARTs from the naïve repertoire are also captured to some extent.

Importantly, CD39 seems to be the parameter that best differentiates the PBMC NART population from the VART population (pre- $p = 0.007$, post-treatment $p = 0.018$; Fig. 4h). The CD39 expression is particularly evident for NARTs, both pre and post-treatment, in the patients with disease control (pre-$p = 0.013$, post-treatment $p = 0.041$; Fig. 4i), indicating recent antigen exposure for this group of T cells. Previously, CD39 has been seen to differentiate tumor specific CD8[+] tumor infiltrating lymphocytes (TILs) from bystander TILs[27]. Interestingly, PD patient #7577 expresses higher frequencies of Ki67[+] cells than the remaining PD patients, but with few CD39[+] NARTs detected both pre- and post-treatment, suggesting lack of antigen recognition for such NARTs, despite proliferation following ICB (Supplementary Fig. 6c, d).

Examples of the above NART subpopulations at three weeks post-treatment is shown in Fig. 4j–p (patient #2389, Best RECIST 1.1 CR). For patients with long-term clinical response, we further evaluated PBMCs at a late time-point post-treatment (231+ weeks post-treatment, $n = 6$). The majority of these patients experienced an initial burst in the frequencies of Ki67[+], PD-1[+] and CD39[+] subpopulations, which declined to pre-treatment levels at the late time-point evaluated (Fig. 4q).

Taken together, a proliferative burst of NARTs is observed following a single dose of PD-L1 blockade, which has been noted previously for bulk CD8 T cells[13]. During this burst, NARTs in patients with disease control tend to be in a Ki67[+] state, and mostly of a CD45RA[−], PD-1[+] phenotype, favoring CD27 expression over CD57. Furthermore, this NART subpopulation can be identified in PBMCs based on CD39 expression.

**Eluted ligand rank-score is a key correlate for neoepitope recognition by CD8[+] T cells.** It is of key interest to precisely predict which tumor neoepitopes are recognized by T cells. Hence, we evaluated a number of features that may impact the likelihood for T cell recognition. T cell recognized neoepitopes had lower percentile rank both related to EL%Rank ($p = 0.0016$) and binding affinity prediction (BA%Rank, $p < 0.0001$), whereas neoepitope-related gene expression level did not differ between T cell recognized and non-recognized neoepitopes ($p = 0.73$; Fig. 5a–c). An enrichment of T cell recognized neoepitopes was observed for predicted neoepitopes with EL%Rank < 0.5 and expression level >2 TPM ($p < 0.001$; Fig. 5d), indicating that gene-expression level in combination with EL is relevant in predicting immunogenic neoepitopes in this cohort. Furthermore, there were no differences in improved- or conserved HLA-binders, as both where equally represented in the T cell recognized fraction ($p = 0.19$, Fig. 5e)[28]. Interestingly, we did not observe improved T cell recognition of neopeptides derived from certain classes of mutations ($p = NS$

Fig. 5f). However, a substantial pool of predicted neopeptides derived from non-missense mutations elicited a T cell response, which was also seen in patients with renal cell carcinoma (RCC)[29]. Such non-missense mutations where unevenly distributed, but present in the majority of evaluated patients (Fig. 5g)

Recently, clonality of predicted neopeptides has been associated with T cell immune reactivity following ICB, with peptides derived from clonal mutations dominating the elicited T cell responses following PD-1- and CTLA-4 blockade[30,31]. To define favorable characteristics of CD8[+] T cell-reactive neoepitopes, we investigated their clonality, gene origin, expression level, and HLA-binding affinity. Peptides derived from clonal mutations make up the majority of neopeptides included in the libraries (5,756 of 6,237; 92%), similar to what has previously been seen in NSCLC[30]. Of 6,237 neopeptides screened in this cohort, 148 unique neopeptides were observed to elicit a T cell response. Of these, 143 neopeptides were derived from clonal mutations, with two originating from cancer driver genes (GPC3 and MAML2). No difference in T cell recognition was observed towards clonal or non-clonal neoepitopes ($p = 0.4$), but that may be due to the large fraction of clonal mutations observed in this cohort. Also, no preference for cancer driver genes amongst recognized peptides was observed ($p = 0.066$) and NARTs recognized peptides from a multitude of non-classical cancer driver genes in this cohort ($n = 120$ genes; Fig. 5h). Together, these results point to the importance of HLA-peptide binding affinity in successfully predicting immunogenic neoepitopes as well as a potential influence of neoantigen expression.

**Pre-treatment TME mRNA gene signatures are associated with post-treatment NART repertoire and phenotypic characteristics.** Having established the characteristics of NART responses, we subsequently analyzed mRNA expression patterns in pre-treatment tumors to evaluate the potential determinants in the tumor microenvironment (TME) driving post-treatment NART response. To better understand the composition of the TME, we applied differential expression analysis (DEA) to determine overly expressed genes and used Microenvironment Population Counter (MCP) to estimate immune cell population abundancy in the TME.

From the pre-treatment RNA-seq data, differentially expressed genes were assessed based on the number of detected NART responses at three weeks post-treatment (high >= median no. of responses of three; Fig. 6a). Interestingly we observed a strong clustering of patients based on the TME gene expression patterns when evaluated based on the differentially expressed genes - patients with high NART response numbers tended to have a higher mRNA expression of genes such as CD3D, PPARG, and TNFSF15, involved in T cell activation and differentiation[32–36], indicating a T cell stimulating pre-treatment TME in patients with high number of post-treatment NART responses. In

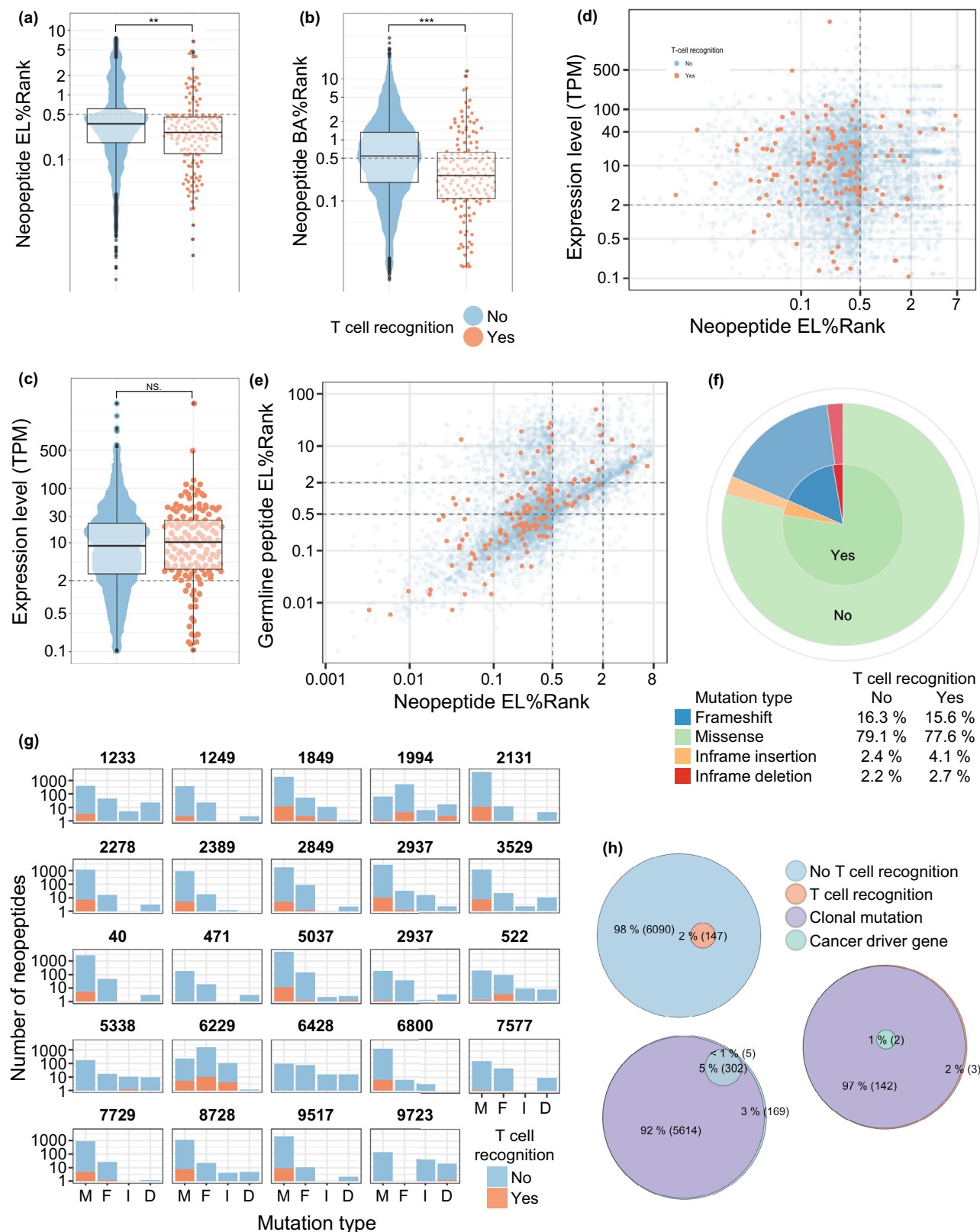

contrast, for patients with lower number of NART responses, we observed high expression of FN1, ITGA5 and COL3A1, which is correlated with poor survival for cancer patients[37–39]. Additionally, CXCL5, involved in angiogenesis[40,41], was increasingly expressed both across patients with low number of NART responses and PD patients. Furthermore, for patients with high NART responses post-treatment, the gene ontology (GO)

enrichment analysis of the DEA results showed pre-treatment enrichment of gene sets associated with the TCR complex and antigen presentation, potentially facilitating an improved NART response post-treatment (Fig. 6b, c).

Applying the MCP-counter method, patients with a high number of detected NART responses post-treatment had a higher expression of genes that together define various subtypes of immune cells,

**Fig. 5 Molecular characteristics of T cell neoepitopes. a–c** EL%Rank score, BA%Rank score and expression level of predicted neopeptides, grouped according to T cell recognition (Yes, $n = 147$ neoepitopes/No, $n = 6,090$ neoepitopes). Dotted lines indicate split for groups in z-test, applied in Fig. 5d. **d** Expression levels of neopeptide-origin genes and predicted EL%Rank score of neopeptides. Colored according to T cell recognition of neoepitopes. **e** Peptide EL%Rank scores of wild type (normal) and mutant neopeptides, colored according to T cell recognition. **f** Proportions of neoepitopes eliciting T cell recognition or not, grouped according to neopeptide-induced type of mutation. **g** Distribution of predicted neopeptides and neoepitope recognition, grouped according the type of peptide-inducing mutation across patients (M = Missense, F = Frameshift, I = Inframe insertion, D = Inframe deletion). **h** Venn-diagram of non- and T cell recognized predicted neopeptides and if they are derived from clonal mutations or cancer-driver genes. For (**a–c**) groups were compared using non-paired *t*-test and data is presented as boxplot ranged by 25th and 75th quartiles with median values, and whiskers as ±largest/smallest value within upper/lower quartile ± 1.5 IQR. NS. Not Significant, \*$p < 0.05$, \*\*$p < 0.01$, \*\*\*$p < 0.001$. Source data are provided as a Source Data file.

including bulk T cells ($p = 0.042$) and with a trend for CD8 T cells ($p = 0.066$), while exhibiting lower levels of fibroblasts, indicating a less inhibitory TME[42] ($p = 0.0005$; Fig. 6d–h). Together, pre-treatment TME patterns are associated with the post-treatment NART response repertoire, with a pre-treatment T cell activating mRNA signature in the tumor within patients that tended to raise a broader NART response post-treatment.

## Discussion
This study utilized a high-throughput screening approach to serially interrogate CD8 T cell recognition of patient-specific neopeptides predicted from the pre-treatment tumor mutagenome in the peripheral blood of 24 patients with mUC treated with anti-PD-L1-therapy. Several findings are important. First, we observed an increase in NART responses from pre-treatment to three weeks post-treatment in patients with disease control. At this time point, the overall neoepitope recognition breadth, not the estimated frequency of such CD8 T cell populations, was associated with clinical radiographic response. This may reflect the finding that the majority of NART responses are low frequent. Second, phenotypic characterization of NARTs revealed an association between Ki-67+ PD-1+ NARTs three weeks post-treatment and clinical outcome. Third, CD39 was expressed on a higher fraction of the NARTs compared with VARTs in the blood, suggesting this marker could be used to identify anti-tumor T cells. The same difference was not observed in bulk CD8+ T cells, which may include additional tumor-antigen specific T cells not captured in our screening for neoepitope recognition. Fourth, TME mRNA expression patterns pre-treatment were associated with increased NART responses three weeks post-treatment. Finally, in silico modelling of neoepitope prediction partially recapitulated the T cell recognition of NARTs, demonstrating peptide HLA-binding, and mutation gene-expression to affect neoepitope T cell recognition.

The apparent kinetics of NART responses detected in our study are consistent with a growing body of literature indicating that ICB rapidly induces an immunological T cell response in the peripheral blood, where multiple reports describe early peripheral T cell turnover, expansion and activation after ICB initiation within 7–21 days[12–17,19,43]. Both effector T cell expansion[14] and Ki67+ PD-1+ CD8 T cell increase[13] in the periphery three weeks post-treatment have been linked to clinical outcome to ICB. Our findings provide novel insight, as they shed light on the specificities and temporal dynamics of neoepitope-recognizing CD8 T cell responses through the detection and quantification of circulating NARTs following ICB. Yet whether emerging NARTs are truly de-novo primed or present pre-treatment at sub-detection levels in periphery cannot be deduced, although recent studies have suggested ICB-induced de-novo recruitment of naïve tumor-specific T cells from the lymph nodes[44].

It should also be noted that the median time to response for patients included in this cohort is 2.1 months (95% CI 2.0–2.2)[5] consistent with the time of the first radiographical disease assessment, comparable to other ICB clinical trials in mUC

patients (median time to response 1.4–2.1 months)[45–50]. Consequently, any earlier tumor reduction prior to first scan is not measured, but the rapid time to response indicates that clinical response to therapy occurs early. Hence, these observations are consistent with our findings of early proliferation, as most response to therapy are captured at 9 weeks post-treatment.

Furthermore, interrogating NARTs revealed kinetics that were not visible from investigating bulk TCR-seq. These NART responses were personal to each patient under study and persisted, albeit at lower levels than the initial 3 weeks peak, for over 5 years post-treatment initiation in some long-term responders. NART responses shared between patients were not identified in this study.

The ability to in silico model the T cell neoantigen responses induced in vivo, as well as the clinical relevance of in silico modeling of antigenic diversity beyond association of TMB and neopeptide prediction to clinical outcome, are areas of active investigation. The importance of gene expression level for neoantigen quality is previously described, yet thresholds remain undefined[51], in contrast to the established importance of HLA-I binding affinity[28]. We observed that the majority of NART populations recognized neopeptides with EL%Rank <0.5 and expression level >2 TPM, and that analyses based on only these responses also were associated with clinical outcome (Supplementary Fig. 4b–d). However, a multitude of NART responses towards neopeptides outside these specifications was detected, and could likewise serve as important targets for anti-tumor immunity. Although the combined EL%Rank scores and expression level provide the best parameters for neopeptide T cell reactivity here, these results also suggest that future neoepitope prediction and selection should include additional parameters, such as tumor immunogenicity, immune priming, and peptide sequence similarity to known self- and infection-derived antigens, as incorporated in recently described neoantigen fitness models[52–56]. At the initiation of study, neopeptides were selected only based on HLA-binding affinity and expression level >0.1 TPM, limiting potential bias in prediction and selection of immunogenic neoepitopes. Recently, selection based on such key characteristics were supported, but also revealed the persistent challenge in fully defining the parameters critical for high accuracy neoepitope prediction[57]. Lastly, the results propose that predicted neopeptides outside these current thresholds should be included in future NART screenings.

Interestingly, we observed that mRNA-expression patterns in pre-treatment tumors differ between patients mounting a wider post-treatment NART response and those that do not. In particular, high T cell infiltration seems to be important for the generation of NARTs. Although needing validation, our results also suggest that pre-treatment TME mRNA gene expression patterns may also be useful when predicting NART responses. Also of interest, clinical outcome may potentially be driven by a combined favorable pre-treatment TME and induced post-treatment NART repertoire, requiring further interrogation.

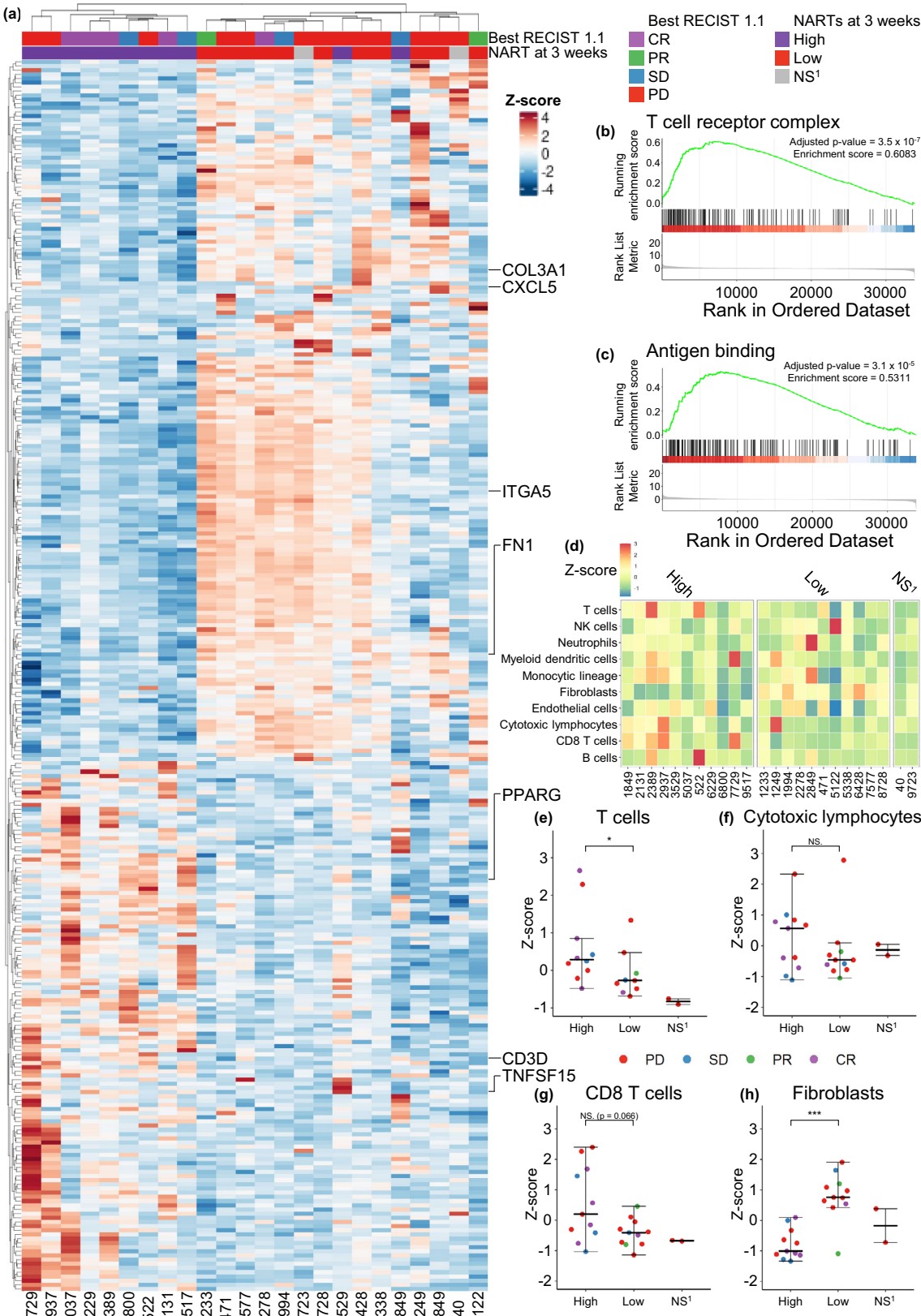

Phenotypic characterization of NARTs allowed for important observations in the peripheral blood that have heretofore been made mostly in the tumor microenvironment. Recently, in NSCLC patients, activated progenitor-like (TCF-1+ PD-1+) T cells with proliferative capacities have been observed, highlighting the likely importance of this cell type to the anti-tumor response[58]. Our analyses revealed that a similar PD-1+ Ki67+ NART population was detected post-treatment primarily in patients who derived benefit from therapy. Although phenotypic changes to some extent seemed to be antigen-independent, our results still suggest that the reactivity and proliferative tendencies of NARTs in the peripheral blood contribute to tumor clearance

**Fig. 6 Transcriptomic analysis of TME related to the level of NARTs post therapy. a** Differentially expressed genes ($n = 295$) from DEA of all patient genes, related to high versus low NART responses at three weeks post-treatment. **b, c** Two significant gene sets from the GO enrichment analysis, antigen binding and T cell receptor complex, respectively. **d** Heatmap of immunological cell signatures across patients, grouped by high or low number of NART responses at 3-week post-treatment, and (**e–h**) associations towards high ($n = 11$ patients) or low number ($n = 11$ patients) of NART responses at 3-week post-treatment for (**e**) T cells, **f** CTLs, **g** CD8 T cells, and (**h**) Fibroblasts. No three weeks post-treatment PBMCs were available from patients #40 and #9723, hence these patients were not included in the analyses or comparisons. For (**a**) and (**d–h**) NS[1] denotes patients not screened at three weeks post-treatment due to sample unavailability. For e)-h) groups were compared using non-parametric two-sided Mann–Whitney test and data is presented as median values ± largest/smallest value within upper/lower quartile ± 1.5 IQR. NS. Not Significant, *$p < 0.05$, **$p < 0.01$, ***$p < 0.001$. Source data are provided as a Source Data file.

and that an activated T cell profile is associated to favorable clinical outcome. Further, in lung and colorectal tumors, CD39 expression has been utilized to distinguish between bystander and NARTs in the microenvironment[27]. We demonstrate here that CD39 expression may also helpful identify NARTs from bystander T cells in the blood without the need to procure and immediately process fresh tumor tissue.

There are notable limitations to our study. First, the sample size was small. However, the screening of the 24 patients under study was unselected and comprehensive, with 200-587 neopeptides included per patient yielding a total of 6237 neopeptides. Patient samples were also longitudinally collected and screened, which allowed for detection of NART responses for up to 5+ years post-treatment in some long-term responders. Second, we did not evaluate the contribution of MHC-II restricted peptides and the role of CD4 T cells, which were recently shown to mediate and drive anti-tumor cytotoxicity and immunogenicity following ICB, also in UC[59–61]. Finally, we did not perform on-treatment biopsies and were therefore unable to evaluate NART trafficking and tumor immunoediting during treatment.

Of additional interest, the T cell neoepitope recognition profile of two patients with partial response to therapy differed significantly from the other patients with disease control. Patient #5122 (PFS/OS = 1932, best RECIST 1.1 PR) differed from other patients with long-term progression-free survival in that the number of NART responses decreased from baseline and maintained no detectable NART responses throughout treatment. However, the patient was observed to harbor a *PDL1* gene amplification, which has been associated with positive outcomes following ICB treatment[62]. Patient #1233 displayed NART dynamics and phenotype as well as a pre-treatment TME mRNA expression pattern associated with PD in the larger cohort; thus, no such characteristics explain the apparent clinical response. These cases indicate the complexity of clinical response and resistance to ICB.

Despite these limitations, our observations add to the current understanding of anti-tumor immunity induced by ICB and suggest that important insights into NART dynamics can be made from pre-treatment archival tumors and post-treatment peripheral blood interrogation alone; NARTs are indeed detectable in the periphery and to a higher degree among patients that benefit from therapy. These findings warrant further investigation, to both improve ICB clinical outcome prediction and to investigate the mechanistic underpinnings of ICB, whether pre-existing NARTs rise to a detectable levels[63] or if NARTs are recruited and activated as a result of ICB[44,64,65].

## Methods and Materials

**Study design and participant samples.** Patients had mUC and were treated with atezolizumab 1200 mg intravenously (IV) every 21 days at Memorial Sloan Kettering Cancer Center as part of the IMVigor210 trial[5]. Cross sectional imaging was performed every 9 weeks for the first 12 months following cycle 1 day followed by every 12 weeks thereafter. Best overall response was determined by radiologic assessment of response, using RECIST version 1.1.

In patients in whom clinical progression was determined based on symptoms and decline and functional status, this determination superseded radiologic categorization. Patients that were only screened at baseline were classified as with progressive disease. All patients provided written informed consent to both the IMvigor 210 trial and an Institution Review Board-approved biospecimen protocol permitting tissue and blood collection, sequencing, and correlative studies.

Twenty-four patients included in a previously published multi-omic analysis[19] were the subject of this study (Supplementary Fig. 1). PFS and OS were updated for this cohort. Patients with PFS > 6 months ($n = 9$; PFS < 6 months $n = 15$) were stated as having DCB from treatment, with other outcomes data in the supplement (Supplementary Table 2). Patient formalin-fixed paraffin-embedded (FFPE) tumor and PBMC samples were obtained and prepared as previously described[19]. Blood samples were drawn from patients prior to IV infusion on the day of treatment, pre-treatment and during treatment, and PBMCs were isolated and cryopreserved at −150 °C in Human Serum Albumin (HSA)/10% DMSO until analysis.

HD samples were collected by approval of the local Scientific Ethics Committee, with donor written informed consent obtained according to the Declaration of Helsinki. HD blood samples were obtained from the blood bank at Rigshospitalet, Copenhagen, Denmark. All samples were obtained anonymously. PBMCs from HDs were obtained from whole blood by density centrifugation on Lymphoprep (Axis-Shield PoC, cat# 1114544) in Leucosep tubes (Greiner Bio-One, cat# 227288) and cryopreserved at −150 °C in fetal calf serum (FCS, Gibco, cat#10500064) + 10% dimethyl sulfoxide (DMSO, Sigma-Aldrich, cat#C6164).

**WES, RNA- and TCR-seq, HLA typing and Next-Generation Sequencing data processing.** Patient WES and RNAseq data, HLA typing and TCRβ CDR3 region amplification was acquired as described as part of a previous study on the patient cohort[19]. In total, 22 patients had tumor and PBMC material of sufficient quality for both WES, RNA-seq, TCRβ-seq and minimum one pretreatment PBMC sample, as tumor TCR analysis was not performed on patient tumor samples from patients #522 and #6800 due to failed sequencing quality control.

Novel for this study, raw FASTQ files from WES and RNAseq were analyzed in the following manner. First, both data sets were pre-processed for quality using Trim Galore version 0.4.0[66], which combines the functions of Cutadapt[67] and FastQC 0.11.2:[68] trimming the reads below an average Phred score of 20 (default value), cutting out standard adaptors such as those from Illumina, and running FastQC to evaluate data quality. Variant calling was performed following the Genome Analysis Toolkit (GATK) best practice guidelines for somatic variant detection[69]. Reads were aligned to the human genome (GRCh38) using the Burrows-Wheeler Aligner[70] version 0.7.15:q with default mem options and with a reading group provided for each sample for compatibility with the following steps. Duplicate reads were marked using Picard-tools version 2.9.1 MarkDuplicates. Base recalibration was performed with GATK version 3.7 to reduce false-positive variant

calls. SNV and indel calls were made using GATK version 3.8's build in a version of MuTect2[71] designed to call variants, both SNVs and indels, from matched tumor and normal samples. Kallisto 0.42.1[72] was used to determine the gene expression in transcript per million (TPM) from RNAseq data.

**Neopeptide prediction and selection.** The VCF output files from GATK's MuTect2 was given as input to the neopeptide predictor MuPeXI version 1.1.3[23] together with RNAseq expression values obtained from Kallisto. HLA alleles of each patient were inferred from the WES data using OptiType version 1.2[73] with default settings after filtering the reads aligning to the HLA region with RazerS version 3.4.0[74]. Identified mutations were used to predict 9, 10, and 11 amino acid peptides, sorted according to the EL% Rank score of the mutated neopeptides using NetMHCpan 4.0[24]. All HLA-I-feasible neopeptides with an EL%Rank score <0.5 with expression level >0.1 TPM were selected for peptide synthesis. For patients where the number of HLA-I-feasible predicted neopeptides did not exceed 200, the highest-ranking 200 peptides were selected to constitute the patient neopeptide library.

**Peptides.** All selected mutation derived and virus control peptides were ordered and purchased from Pepscan (Pepscan Presto BV, Lelystad, Netherlands). Peptides were dissolved to 10 mM in DMSO following arrival and stored at −18 °C prior to use.

**MHC monomer production and generation of peptide-MHC complexes.** The production of MHC monomers was performed as previously described[75,76]. In brief, MHC class I heavy chains and human β2m were expressed in E.coli strain BL21(DE3) pLysS (Novagen, cat#69451). Inclusion bodies containing expressed proteins were harvested by washing in detergent buffer and wash buffer and solubilized in 8 M urea buffer (8 M Urea, 50 mM K·HEPES pH 6.5, and 100 μM β-mercaptoethanol). Final purified inclusion bodies were stored at −80 °C until used. MHC class I molecules were obtained by in vitro folding of heavy chain and β2m light chain with respective UV-sensitive ligand in folding buffer (0.1 M Tris pH 8.0, 500 mM L-Arginine-HCl, 2 mM EDTA, 0.5 mM oxidized glutathione and 5 mM reduced glutathione) at 4 °C[77,78] or by using disulfide-stabilized empty MHC I complexes as previously reported[79]. After folding for 3–5 days, folded protein was biotinylated using BirA biotin-protein ligase standard reaction kit (Avidity, LLC- Aurora, Colorado). Finally, biotinylated monomer complexes were purified with size-exclusion column (Waters, BioSuite SEC Column, 125 Å, 13 μm SEC, 21.5 mm × 300 mm) with HPLC (Waters Corporation, USA)) and stored at −80 °C until further use. Specific pMHC complexes were generated by UV-induced peptide exchange[75,77].

**Detection of peptide-MHC specific T cells by DNA barcode-labelled multimers.** Patient-specific libraries of predicted neopeptides and virus control peptides (size 201-589 peptides per patient) were generated as DNA barcode-labelled pMHC multimers as previously described[22]. In short, patient specific neoepitope MHC multimer libraries were generated by multimerizing exchanged pMHC molecules on a PE-labeled polysaccharide-backbone (for neopeptides), and APC-labeled polysaccharide-backbone (for virus-derived epitopes) coupled to DNA barcoded-labeled dextran backbone Thus, a specific peptide is linked to a unique DNA barcode together with a fluorescent label, serving as a tag for the given pMHC. Patient and HD PBMCs were stained with an up-concentrated pool of multimers together with 50 nM dasatinib. Samples screened only for T cell multimer recognition were stained with an antibody mix consisting of CD8-BV480 (BD, cat. #566121, clone RPA-T8, 2 μl), dump channel antibodies (CD4-

FITC (BD, cat. #345768, 1.25 μl), CD14-FITC (BD, cat. #345784, 3.125 μl), CD19-FITC (BD, cat. #345776, 6.25 μl), CD40-FITC (Serotech, cat. #MCA1590F, 2.5 μl), and CD16-FITC (BD, cat. #335035, 1.56 μl)) and a dead cell marker (LIVE/DEAD Fixable Near-IR; Invitrogen, cat. #L10119, 0.1 μl). Multimer binding T cells were sorted as lymphocytes, single, live, CD8$^+$, FITC$^-$ and PE$^+$. Samples screened for T cell multimer recognition and phenotypic characterization were stained with an antibody mix composed of T cell lineage markers (CD3-BV786 (BD, cat. #563799, clone SK7, 5 μl), CD4-BV650 (BD, cat. #563876, 2.5 μl), and CD8-BV480 (BD, cat. #566121, clone RPA-T8, 2 μl)), characterization markers (Ki67-BUV395 (BD, cat. #564071, clone B56, 2.5 μl), 4-1BB-BUV737 (BD, cat. #741867, clone 4B4-1, 2.5 μl), PD1-BV421 (BioLegend, cat. #329920, clone EH12.2H7, 3 μl), CD27-BV605 (BioLegend, cat. #302829, clone O323, 2.5 μl), CD45RA-BV711 (BD, cat. #563733, clone HI100, 2.5 μl), CCR7-FITC (BioLegend, cat. #353215, clone G043H7, 5 μl), Eomes-PerCP-eFlour710 (eBioscience, Thermo Fisher Scientific cat. #46-4877-41, clone WD1928, 2.5 μl), CD39-PE-CF594 (BD, cat. #563678, clone Tu66, 2.5 μl), CD57-PECy7 (BioLegend, cat. #393309, clone QA17A04, 2.5 μl), and GranzymeB-AlexaFluor700 (BioLegend, cat. #372221, clone QA16A02, 1.25 μl)), and a dead cell marker (LIVE/DEAD Fixable Near-IR; Invitrogen, cat. #L10119, 0.1 μl). Multimer binding CD8 + T cells were sorted as lymphocytes, single, live, CD3$^+$, CD8$^+$, CD4$^-$ and either PE$^+$ or APC$^+$ on either FACSAria$^{TM}$ Fusion or FACSMelody$^{TM}$ instruments (BD Biosciences). All sorted T cells were pelleted by centrifugation. From isolated cells and from a stored baseline aliquot of multimer pool (diluted 10,000x in final PCR reaction), the specificities of multimer$^+$ CD8$^+$ T cells were decoded by amplification, subsequent purification using QIAquick PCR Purification kit (Qiagen, cat. #28104), and ultimate sequencing of DNA barcodes at PrimBio Research Institute (PA, USA) using an Ion Torrent PGM 316 or 318 chip (Life Technologies). Sequencing data were processed by the software package Barracoda, available online at (https://services.healthtech.dtu.dk/service.php?Barracoda-1.8). The tool identifies the DNA barcodes annotated for a given experiment, assigns a sample ID and pMHC specificity to each DNA barcode, and counts the total number of reads and clonally reduced reads for each peptide-MHC-associated DNA barcode. Log$_2$FC in read counts mapped to a given sample relative to the mean read counts mapped to triplicate baseline samples are estimated using normalization factors determined by the trimmed mean of M-values method. FDRs were estimated using the Benjamini–Hochberg method. A minimum read count fraction of 0.1% for a given DNA barcode of the total DNA barcode number in that given sample was set as threshold to avoid false-positive detection of T cell responses due to low number of reads in the baseline samples. DNA barcodes with FDR <0.1% (corresponding to p < 0.001), read count fraction > (total read count for full barcode library/barcode library size), Log$_2$FC >2 over the input values for the total pMHC library, and CD8+ T cell estimated frequency of >0.01 % were considered to be true T cell responses, based on previous studies[22,29,80].

**Detection of peptide-MHC specific T cells by fluorescently-labelled tetramers.** For selected neopeptides, pMHC tetramers were generated for staining of neoepitope-specific T cells. Neopeptides were selected based on the observed NART responses from the DNA barcode-labelled multimer screening. Following the observed increase in NART responses at 3 weeks post-treatment, NART responses in 3-week post-treatment PBMC samples were interrogated wherever sufficient patient material remained, otherwise 9-week post-treatment samples were analyzed. Single-fluorochrome pMHC specificity tetramers using were generated as described in detail previously[81,82], using a library of streptavidin (SA)-conjugated flourochromes consisting of PE-SA (BioLegend, cat. #405204), APC-SA (BioLegend,

cat. #405207), BV421-SA (BD, cat. #563259), PE-Cy7-SA (BD, cat. #557598), BV605-SA (BD, cat. #563260), PE-CF594-SA (BD, cat. #562284), BV650-SA (BD, cat. #563855), BUV395-SA (BD, cat. #564176). Up to eight patient-specific pMHC tetramers per sample were investigated. PBMC samples were stained with respective library of pMHC tetramers and with an antibody mix consisting of CD8-BV480, dump channel antibodies and a dead cell marker, as above. Tetramer-specific T cells analyzed as lymphocytes, single, live, CD8$^+$, FITC$^-$ and tetramer$^+$ cells. Due to staining strategy, tetramer$^+$ cells were gated by being CD8$^+$.

**Analytical processing of detected T cell responses**. For each patient, patient PBMC samples at timepoints pre-, during- and post-treatment were stained with the respective patient pMHC multimer library and relevant fluorescent antibodies. Concurrent with patient PBMC samples, PBMCs from HDs, HLA-matched with the respective patient as best possible, were also stained with the patient's pMHC multimer library and CD8 T cell subset identification antibodies (one to three HD samples per staining, median = 2). Presence of NARTs was determined based on the enrichment in barcode reads for a given neoepitope, visualized as the Log$_2$FC for each pMHC specificity in each patient sample, longitudinally evaluated over the course of treatment (Fig. 2a). For all patient and HD sample screenings, samples were stained with the entire multimer library. Predicted for and included in the screenings, the library of neopeptides presented on HLAs C0202 and C0501 (n = 515) were subsequently excluded from downstream analysis due to observations of substantial unspecific binding of these HLAs to Killer-cell immunoglobulin-like receptor (KIR). Furthermore, to avoid signals from potential pMHC-elements with unspecific binding, all HLA-matching patient-derived NART responses detected in HD samples were excluded from the pool of patient T cell responses, if the given neopeptide response was detected across all samples for a given patient. Of 6237 screened neopeptides in all patient and HD PBMC samples, 28 neopeptides generated background signals, and thus was excluded from the final library of NART responses.

For any pMHC-coupled DNA barcode in a sample, an estimated frequency of each pMHC-recognizing CD8$^+$ T cell population was estimated based on the read count fraction of the given DNA barcode out of the total fraction of T cells binding the pMHC multimer pool. Estimated frequencies for all NART responses were summed-up for each patient and timepoint to determine the total frequency of NARTs, i.e. sum of estimated frequencies.

**Analytical processing of phenotyping data**. Flow cytometry results were analyzed using the FlowJo v10 software (TreeStar, Inc.)[83]. For UMAP dimensionality reduction, 3000 representative live, CD3$^+$, CD4$^-$, bulk CD8$^+$ T cells from patient samples pre- and 3 weeks post-treatment were concatenated (n = 28) and projected using the UMAP plugin in FlowJo[84]. UMAP was run by selecting the parameters for Eomes, GzmB, CD27, CD57, CD45RA, CCR7, CD39, PD1, and Ki67. 41BB-BUV737 was excluded from further analysis due to significant spectral overlap from CD45RA-BV711. For the FlowSOM algorithm for unsupervised clustering[85], 15 clusters were selected with otherwise default settings.

**Differential expression analysis and microenvironment cell populations-counter**. Differential expression analysis is performed with all genes where the output from Kallisto version 0.42.1 was used as input to DeSeq2[86] version 1.26.0 from Bioc-Manger in R version 3.6.1 with default option. The median number of detected NART responses at the given time point was used to split the cohort in high versus low number of NART responses. Log-fold change >1 and <−1 together with an adjusted

p-value < 0.05 was used as threshold for over- and under-expressed genes for the analysis. The heatmap illustrations were generated with ComplexHeatmap from Bioconductor[87]. The GO enrichment analysis is developed using R version 4.1.1 with the built in packages; enrichplot version 1.13.2[88], clusterProfier version 4.0.5[89] with Benjamin Hochberg at p value adjustment. Cell populations abundancy was estimated from bulk RNA sequencing data using Microenvironment Cell Populations-counter (MCP-counter)[90]. The expression matrix obtained from Kallisto was fed as input to ebecht/MCPcounter from GitHub in R version 4.0.2 with Hugo-symbols as feature Type.

**Neopeptide clonality**. Allele copy number, purity and ploidy were found using Sequenza version 3.0[91]. As input, bam files from normal and tumor were given to Sequenza-utils version 3.0 bam2seqz with CRCh38 as reference followed by Sequenza seqz_binding. To run the Sequenza copynumber call with CRCh38, the R packages copynumber[92] with minor modifications from Shixiang/copynumber[93] was applied. Sequenza results were generated with the Sequenza packages in R version (3.6.1) and copynumber information from Sequenza were merged with mutations file from Mutect2 and used as input to PyClone. To locate clonal mutations, PyClone version (0.13.0)[94] was applied with the best estimated cellularity given from Sequenza, and max cluster of 10 and minimum size of 0 to yield all possible mutations. Afterwards, clonal mutations were filtered with a cluster size of minimum 5 and cellularity of minimum 90.

**Figures and statistical analysis**. Figure 1a and 2a were created in Biorender. Graphs in Fig. 1b–d, 2b, c, 3a–i, 4d–i + q, 5a–h, 6d–h, Supplementary Fig. 2a–c, 4a–e +m, 5a–e and 6j–p were generated using the *ggplot2* package in R v3.6.1 and v4.0.2. For Fig. 3f–g and Supplementary Fig. 4a, to facilitate plotting on logarithmic scale, in cases where SEF was 0 due to no detected T cell responses, 0.001% was added to SEF. Groups in Fig. 1c–d, 3a, b, d, f, h, 4d–g, 5g and 6e–h, and Supplementary Fig. 4a–c + e, 5a–c +e and 6b +e–g were compared using non-parametric two-sided Mann–Whitney test, in Fig. 3c, e, g, i and 4h–i, and Supplementary Fig. 4d and 5d using Kruskal–Wallis Dunn's multiple comparison test, and in Fig. 5a–c using non-paired t-test. Mann–Whitney and t-test conducted using the *ggsignif* package[95] and Dunn's test using the *rstatix* package[96]. Proportion test (z-test)[97] was applied for Fig. 4b and 5d–f + h. The *eulerr* package[98] was applied for Fig. 5h. Figure 6a was generated from the *ComplexHeatmap* and Fig. 6b–c from *clusterProfiler* and *enrichplot* packages in R. All survival plots (Supplementary Fig. 4f-h) were made using Kaplan Meyer-curves and hazard ratios with the *survival* and *surviminer* packages in R[99]. Figure 4a–c + j–p and Supplementary Fig. 3 and 6a were generated from FlowJo v10. For all figures; NS. Not Significant, *p < 0.05, **p < 0.01, ***p < 0.001, ****p < 0.0001.

**Reporting summary**. Further information on research design is available in the Nature Research Reporting Summary linked to this article.

## Data availability

All WES and RNAseq data is available upon application at dbGaP at https://www.ncbi.nlm.nih.gov/projects/gap/cgi-bin/study.cgi?study_id=phs001743.v1.p1. GRCh38 reference genome is available at https://www.ncbi.nlm.nih.gov/assembly/GCF_000001405.39. The source data presented in figures are provided as Source Data file. All other relevant data are available from the authors upon request. Source data are provided with this paper.

## Code availability

The applied codes are embedded into MuPeXI and Barracoda, both publicly available tools, and can be reached and accessed at https://github.com/ambj/MuPeXI and https://services.healthtech.dtu.dk/service.php?Barracoda-1.8, respectively. All other data and R

scripts to reproduce figures can be obtained from the corresponding author upon request.

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

## Acknowledgements

We thank all donors and patients for participating in the study; B. Rotbøl and A.F. Løye for technical assistance handling the flow cytometry instruments and PBMC samples; C. Heeke for assistance in pMHC multimer gathering; and R. Zappasodi, B. Greenbaum, D. Aggen, M. Hellmann, and V. Balachandran for fruitful discussions of the data. The research was funded in part by the Ludwig Center for Cancer Research; the NIH/NCI Cancer Center Support Grant P30 CA008748, the K12CA184746-01A1 (S.A.F.); Ludwig Collaborative and Swim Across America Laboratory (MSKCC, New York, NY 10065, USA); Parker Institute for Cancer Immunotherapy (MSKCC, New York, NY 10065, USA); Department of Medicine (MSKCC, New York, NY 10065, USA); Weill Cornel Medicine (MSKCC, New York, NY 10065, USA); and the Independent Research Fund Denmark (J.S.H.).

## Author contributions

J.S.H. designed and performed experiments, analyzed the data, generated figures, and wrote the manuscript; S.A.F. collected patient material, analyzed the data, provided clinical evaluation, and wrote the manuscript; A.B. and K.K.M. conducted all bioinformatics analyses and generated figures; A.M.B. predicted neoepitopes; J.R. designed phenotypic panels and experiments; C.M. and A.R. managed patient material; P.W., H.A.A., and G.I. collected patient material and provided clinical evaluation; T.T. produced HLA-I complexes; A.K.B. provided technical guidance; N.O.H. selected neopeptides; S.D.W. discussed the data, A.S. provided WES-, RNA-seq- and TCRb-seq-data; T.M. and J.D.W. supervised the study and discussed the data; M.N. designed the in silico platforms and supervised neopeptide prediction; J.E.R. supervised the patient material collection and clinical evaluation, and discussed the data; D.F.B. conceived the concept, supervised the patient material collection and clinical evaluation, discussed the data, and supported funding; S.R.H. conceived the concept, supervised the study, discussed the data, supported funding, and wrote the manuscript. All authors reviewed and approved the manuscript.

## Competing interests

SAF has received research support from AstraZeneca, Genentech/Roche, is a consultant/advisory board member for Merck, and owns stock in Urogen, Allogene Therapeutics, Neogene Therapeutics, Kronos Bio, and Inconovir. He is also supported by the National Cancer Institute K12CA184746-01A1 grant and the Bochner-Fleisher Research Scholar in Urologic Oncology Award. HAA is a consultant/advisory board member for Bristol Myers Squibb, EMD Serono, AstraZeneca/MedImmune, and Janssen Biotech. GI has a consulting or advisory role for Bayer, Janssen, and Mirati Therapeutics; and has received research funding from Mirati Therapeutics, Novartis, Debiopharm Group, and Bayer. PW is a consultant for Sellas Life Sciences and Leap Therapeutics. AKB and SRH are co-inventors of the licensed patents for DNA Barcoded MHC-multimers (WO2015185067 and WO2015188839), licensee: Immudex, DK. SDW is supported by the MSK Clinical Scholars T32 (5T32CA009512) and Young Investigator Award from the American Society of Clinical Oncology. AS is an employee of and owns stock in Merck and Co. TM is a consultant for Leap Therapeutics, Immunos Therapeutics and Pfizer, and co-founder of Imvaq therapeutics; has equity in Imvaq therapeutics; has received research support from Bristol Myers Squibb, Surface Oncology, Kyn Therapeutics, Infinity Pharmaceuticals, Peregrine Pharmaceuticals, Adaptive Biotechnologies, Leap Therapeutics and

Aprea; and is inventor on patent applications related to work on oncolytic viral therapy, alphavirus-based vaccines, neo-antigen modeling, CD40, GITR, OX40, PD-1 and CTLA-4. JDW is a consultant for Amgen, Apricity, Arsenal IO, Ascentage Pharma, AstraZeneca, Astellas, Boehringer Ingelheim, Bristol Myers Squibb, Chugai, Dragonfly, F Star, Eli Lilly, Georgiamune, Imvaq, Merck, Polynoma, Psioxus, Recepta, Trieza, Truvax, Sellas, and Werewolf Therapeutics; has equity in Tizona Pharmaceuticals, Imvaq, Beigene, Linneaus, Apricity, Arsenal IO, and Georgiamune; and has received research support from Bristol Myers Squibb and Sephora. JDW is also the co-inventor of the following licensed patents: Xenogeneic DNA Vaccines (USPTO, US7556805), licensee: Merial; Myeloid-derived suppressor cell (MDSC) assay (EPO, PCT/US2013/027475), licensee: Serametrix; Anti-PD1 Antibody (USPTO, US10323091), licensee: Agensus; Anti-CTLA4 antibodies (USPTO, US10144779), licensee: Agensus; Anti-GITR antibodies and methods of use thereof (USPTO, US10155818/US10280226), licensee: Agenus/Incyte. JER holds stock and other ownership interests in Illumina; has received honoraria from AstraZeneca, Bristol-Myers Squibb, Chugai Pharma, Clinical Care Options, Clinical Mind, Intellisphere, Medscape, Peerview, Research To Practice, UpToDate, and Vindico; has a consulting or advisory role for Adicet Bio, Agensys, Astellas Pharma, AstraZeneca/MedImmune, Bayer, BioClin Therapeutics, Bristol-Myers Squibb, EMD Serono, Fortress Biotech, GlaxoSmithKline, Inovio Pharmaceuticals, Janssen Oncology, Lilly, Merck, Pharmacyclics, QED Therapeutics, Roche/Genentech, Seattle Genetics, Sensei Biotherapeutics, and Western Oncolytics; has received institutional research funding from Astellas Pharma, AstraZeneca, Bayer, Genentech/Roche, Incyte, Jounce Therapeutics, Mirati Therapeutics, Novartis, QED Therapeutics, Seattle Genetics, and Viralytics; holds an institutional interest in a patent for a predictor of platinum sensitivity; and has received travel and accommodation expenses from Bristol-Myers Squibb and Genentech/Roche. DFB is a consultant for Bristol Myers Squibb, Merck, Genentech-Roche, Astra-Zeneca, and Pfizer; and has received research support from Merck, Genentech-Roche, Astra-Zeneca, Novartis, and Bristol-Myers Squibb. SRH is the co-founder of Tetramershop and PokeACell and co-inventor of the licensed patent for Combination encoding of MHC multimers (EP2088/009356), licensee: Sanquin, NL. The remaining authors declare no conflicts of interest.
