## [Peer Review File · Nature Communications]

Neoantigen-specific CD8 T cell responses in the peripheral blood following PD-L1 blockade might predict therapy outcome in metastatic urothelial carcinomaREVIEWER COMMENTS

Reviewer #1 (Remarks to the Author):

In this manuscript by Holm et al, the authors have investigated the breadth of NARTs in a cohort of 24 patients with mUC treated with PD-L1-blockade using DNA barcode-labelled pMHC multimers. The authors identified an increase in the numbers of NARTs in patients with disease control after 3 weeks of treatment and have phenotypically profiled NARTs.

The manuscript reads very well, the study is carefully performed and data are well presented.

The above issues should be addressed:

1) The increase in the number of NARTs in disease control patients is convincing but was it specific to NARTs (as opposed to VARTs)?

2) Along the same line, it is surprising that the effect is limited to 3 weeks. It is key to understand if the detection of NARTs in peripheral blood indicates de novo priming of naive cells or, alternatively, recirculation of pre-existing TILs. It would be interesting to assess the presence of NARTs in tumors pre-treatment and to show high PD1 expression.

3) Why is the increase of NARTs limited to 3 weeks? Are the newly-detected NARTs short-lived effector cells or do they rather (re)infiltrate tumors?

4) Besides the breadth of NARTs, was the overall magnitude of NARTs (sum of their frequencies) a correlate of responses?

5) The authors state that "Peripheral NARTs derived from patients with disease control displayed a PD1+ Ki67+ effector phenotype" but this phenotype does not seem to be specific to NARTs as bulk CD8 T cells seem to show the same profile

6) Overall, although cumbersome, it would be key to demonstrate in responders that newly-detected NARTs can recognize autologous tumors.

I recommend acceptance with major revisions

Reviewer #2 (Remarks to the Author):

Summary: Holm et al. investigated the expansion of neoantigen specific CD8 T cell responses following PD-L1 blockade in blood of patients with metastatic urothelial carcinoma. Using their previously published technique for large-scale detection of antigen-specific T cells using peptide-MHC1 multimers with labeled DNA barcodes, they were able to screen peripheral blood samples pre- and post PD-L1 blockade for neoantigen-reactive CD8 T cells (NARTs). The authors showed that 3 weeks after PD-L1 blockade there was an increase in the breadth, but not magnitude, of neoantigen specific CD8 T cell responses in complete responders but not in non-responders. In terms of T cell quality, responders had increased expression of PD-1 and Ki-67 compared to non-responders. Overall, the correlative observations may be worth exploring further to determine the dynamic changes of NARTs in response to PD-L1 blockade.

Major comments:

(1) The longitudinal sampling of 24 patients with 31 different HLA haplotypes and 200–587 neopeptides per patient is a commendable effort, demonstrating the advantage of this technique.

(2) The abstract states that 'the study provides insights into the mechanisms and kinetics'. However, the data shown here is primarily correlational, understandably due to the nature of characterizing clinical samples. Authors should revise this statement.

(3) In Table 1, the number of T cell neoepitopes detected is counted at the indicated time points. It would be useful to track the identity of neoepitopes over time. For example, were the 5 responses pre-treatment also present in the 13 post-treatment, or are these completely new neoepitopes? This information could be added as a supplementary table.

(4) Based on the number of statistical tests performed in Figure 3 panel C and E, and Figure 4 panel H and I, there should have been a correction for multiple comparisons in the calculation of p-values which is not indicated. Probably the appropriate test is Kruskal Wallis with Dunn's correction.

(5) Figure 6 - The individual gene names in the heatmap are difficult to read, so it is hard for the reader to follow the author in identifying genes correlating with increased NARTs 3 weeks after treatment. What do the CD4 T cell transcripts look like? How was it determined that greater than 3 neoantigen specific CD8 T cell responses is considered a high number? Are there no other gene sets that correlate with increased NARTs or outcome? The correlations used in panels e-g don't seem to correlate well with having either increased NART's or patient outcomes.

Minor comments:

(1) Please include statistics in the figure legends so it will be easier for the reader to evaluate the significance of comparisons.

(2) Lines 52-54 are unclear as to how you selected 200 neo-epitopes for each patient if they didn't have enough neoepitopes that cleared selection criteria.

(3) Line 93-94, should read 'were detected in 18 of 24 patients'.

(4) Line 132-133, triple negative statement is confusing 'neither the absolute SEF nor change in SEF is not associated to clinical outcome' – needs to be reworded for clarification.

(5) Line 144 'was conducted' should be 'were conducted'.

(6) Figure 6a, please bold the genes that are referred to in the text and increase text size so they can be easily seen, or perhaps subset the data to include a smaller set of genes that can be more easily seen.

Reviewer #3 (Remarks to the Author):

Holm et al. present a manuscript on "Neoantigen CD8 T cell recognition is broadened and activated in relation to disease control in the peripheral blood of patients with metastatic urothelial carcinoma following PD-L1 blockade". This study utilize a high-throughput screening approach to serially analyze CD8 T cell recognition of patient-specific neopeptides predicted from the pre-treatment tumor mutagenome in the peripheral blood of 24 patients with metastatic urothelial carcinoma treated with anti-PD-L1-therapy. The authors describe a rapid increase in the number of NART responses from pre-treatment to 3 weeks post-treatment initiation. The authors claim, that the overall neoepitope recognition breadth, not the estimated size of such CD8 T cell populations, was associated with clinical radiographic response. mRNA expression patterns of the tumor microenvironment pre-treatment were associated with increased NART responses.

This is a fascinating approach, that can contribute significant knowledge to the field. However, the interpretation of the data raises several questions:

Major:

- Abstract: „significant increase in the number of neoantigen-reactive CD8+ T cell (NART) responses“ remains unclear what is meant ... broadening of epitope specificity or increase in numbers or increased functionality

- Figure 2b: Why is the NART response so inconsistent over time? I assume the example in the figure was not the worst patient example. 4 out of 6 HLA-A2 restricted responses are detected only at one time point (not even all at one time), 2 at two out of 5 time points. Since T-cell responses are expected to be part of long-term immune responses, this raises questions about the validity of single NART events.

- Flowcytometry based validation of these responses with a detectable population using a single peptide MHC multimer would validate responses.

- Page 5, line 96: "multiple patients" please specify

- Page 6, line 114 and Figure 3: Differences are given at time "3 weeks". Are there differences in NART at other time points? What is the time point of response evaluation? Is it always the same time as the NART are detected? Is there a correlation when response has taken place and the NART detection? Can the authors provide evidence, that a NART detection at 3 weeks is responsible for RECIST criteria at 6 months?

- Flow data are not convincing. Parameters of T cell subpopulations Tem, Tcm, Tscm, Teff would be interesting rather than ki67. Exhaustion markers would be helpful to interpret the data.

Minors:

The manuscript is cumbersome to read. Abbreviations are not consistently used and/or explained. Text in the figures is often too small read it in a print out page.

POINT-TO-POINT RESPONSE TO REVIEWER COMMENTS

Reviewer #1 (Remarks to the Author):

In this manuscript by Holm et al, the authors have investigated the breadth of NARTs in a cohort of 24 patients with mUC treated with PD-L1-blockade using DNA barcode-labelled pMHC multimers. The authors identified an increase in the numbers of NARTs in patients with disease control after 3 weeks of treatment and have phenotypically profiled NARTs. The manuscript reads very well, the study is carefully performed and data are well presented.

The above issues should be addressed:

- 1) The increase in the number of NARTs in disease control patients is convincing but was it specific to NARTs (as opposed to VARTs)?

We have further evaluated the VARTs, and no changes in the VART response repertoire were observed during treatment (included in Supplementary Figure 5), however it should be noted that a substantial smaller library of CEF-derived epitopes was included in the analyses compared to neoepitopes. A remark is now included on page 6.

- 2) Along the same line, it is surprising that the effect is limited to 3 weeks.

It is key to understand if the detection of NARTs in peripheral blood indicates de novo priming of naïve cells or, alternatively, recirculation of pre-existing TILs.

It would be interesting to assess the presence of NARTs in tumors pre-treatment and to show high PD1 expression.

Why is the increase of NARTs limited to 3 weeks? Are the newly-detected NARTs short-lived effector cells or do they rather (re)infiltrate tumors?

We have included a series of references from other studies showing that response to therapy is often seen within the 3 weeks to 1-2 months post-treatment, consistent with observed in our study. Our and others' results indicate that the proliferative burst occurs early and may direct the ensuing clinical response to therapy. However, at this point the destination of the NARTs present at 3 weeks post-treatment and not at 9 weeks is experimentally unknown. It is likely that NARTs at pre-treatment either is present in the periphery at sub detection levels, or is truly de-novo primed and recruited, which we cannot deduce from the current study as our technique does not allow for kinetical tracking of individual NARTs from the lymphatics to the periphery and tumor. However, according to recent studies, naïve tumor-reactive T cells may indeed be recruited to the periphery as a results of PD-L1-blockade. Unfortunately tumor material was not available for further analysis.

These considerations has been further addressed in the discussion of the revised manuscript.

- 3) Besides the breadth of NARTs, was the overall magnitude of NARTs (sum of their frequencies) a correlate of responses?

From our current data the broadening of the NART responses seem to be the key factor associated to clinical outcome of ICB. We don't see any independent effect on the 'size', i.e. the combined

frequencies, of such T cell populations. This may reflect the fact that the majority of responses are relatively low frequent. This consideration has been included in the discussion section.

- 4) The authors state that “Peripheral NARTs derived from patients with disease control displayed a PD1+ Ki67+ effector phenotype” but this phenotype does not seem to be specific to NARTs as bulk CD8 T cells seem to show the same profile.

Indeed, the majority of the phenotype changes observed seems to be antigen independent, and may reflect an intrinsic immune activation in the given patients. However, the activated T cell profile was associated to favorable outcome, and possible required for the ICB effect. This has been further clarified in the revised manuscript.

- 5) Overall, although cumbersome, it would be key to demonstrate in responders that newly-detected NARTs can recognize autologous tumors.

Unfortunately, in this patient cohort, we did not have any access to fresh tumor material. Since all neoepitopes responses are personal, an autologous tumor cell line, or fresh/cryopreserved tumor material is needed to provide measures of tumor reactivity. We agree that it would be highly interesting to do, but not possible in the present cohort.

I recommend acceptance with major revisions.

Reviewer #2 (Remarks to the Author):

Summary: Holm et al. investigated the expansion of neoantigen specific CD8 T cell responses following PD-L1 blockade in blood of patients with metastatic urothelial carcinoma. Using their previously published technique for large-scale detection of antigen-specific T cells using peptide-MHCI multimers with labeled DNA barcodes, they were able to screen peripheral blood samples pre- and post PD-L1 blockade for neoantigen-reactive CD8 T cells (NARTs). The authors showed that 3 weeks after PD-L1 blockade there was an increase in the breadth, but not magnitude, of neoantigen specific CD8 T cell responses in complete responders but not in non-responders. In terms of T cell quality, responders had increased expression of PD-1 and Ki-67 compared to non-responders. Overall, the correlative observations may be worth exploring further to determine the dynamic changes of NARTs in response to PD-L1 blockade.

Major comments:

- 1) The longitudinal sampling of 24 patients with 31 different HLA haplotypes and 200–587 neopeptides per patient is a commendable effort, demonstrating the advantage of this technique.
We appreciate that you acknowledge this effort
- 2) The abstract states that ‘the study provides insights into the mechanisms and kinetics’. However, the data shown here is primarily correlational, understandably due to the nature of characterizing clinical samples. Authors should revise this statement.

The abstract has been rephrased according to reviewer's suggestion.

- 3) In Table 1, the number of T cell neoepitopes detected is counted at the indicated time points. It would be useful to track the identity of neoepitopes over time. For example, were the 5 responses pre-treatment also present in the 13 post-treatment, or are these completely new neoepitopes? This information could be added as a supplementary table.

We have added a Supplementary Table 1 listing all NART responses at all timepoints, facilitating the tracking of individual NART responses throughout analyzed samples.

- 4) Based on the number of statistical tests performed in Figure 3 panel C and E, and Figure 4 panel H and I, there should have been a correction for multiple comparisons in the calculation of p-values which is not indicated. Probably the appropriate test is Kruskal Wallis with Dunn's correction.

Kruskal Wallis Dunn's multiple comparison testing has been performed and statistics updated for all figures with multiple comparisons (Figures 3.c,e,g,i and 4.h-i, and Supplementary Figures 4.d and 5.d).

- 5) Figure 6 - The individual gene names in the heatmap are difficult to read, so it is hard for the reader to follow the author in identifying genes correlating with increased NARTs 3 weeks after treatment.

Figure 6 has been revised to improve readability and identified genes have been singled out. Furthermore the full list of differentially expressed genes has been included as a new Supplementary table 3.

- 6) What do the CD4 T cell transcripts look like?

Although detailed insight into CD4 transcripts would be interesting, using the MCP-counter approach it was not possible to identify specific gene expression patterns related to CD4 cells. Any influence from CD4 T cells would be included under the general 'T cells' compartment, also further quantified between patient clusters in Figure 6.e.

- 7) How was it determined that greater than 3 neoantigen specific CD8 T cell responses is considered a high number?

The median no. of NART responses post-treatment was determined as 3 and thus set as cutoff between high and low number of responses. This has been clarified on page 12.

- 8) Are there no other gene sets that correlate with increased NARTs or outcome? The correlations used in panels e-g don't seem to correlate well with having either increased NART's or patient outcomes.

The gene set displayed in Figure 6.a comprises of all significantly enriched genes. A list of all genes in the set has been included as raw data (supplementary table 3). For the T cell compartment signatures generated from MCP counter, either a significant difference or a trend was observed when comparing NART high to NART low patients (Figure 6.e-h).

Minor comments:

- 9) Please include statistics in the figure legends so it will be easier for the reader to evaluate the significance of comparisons.
Information about the statistical test performed is included in each figure legend. The p values are included in the result section.
- 10) Lines 52-54 are unclear as to how you selected 200 neo-epitopes for each patient if they didn't have enough neoepitopes that cleared selection criteria.
This has been rephrased and specified.
- 11) Line 93-94, should read 'were detected in 18 of 24 patients'.
This section has been revised.
- 12) Line 132-133, triple negative statement is confusing 'neither the absolute SEF nor change in SEF is not associated to clinical outcome' – needs to be reworded for clarification.
The sentence has been rephrased for clarification.
- 13) Line 144 'was conducted' should be 'were conducted'.
Corrected.
- 14) Figure 6a, please bold the genes that are referred to in the text and increase text size so they can be easily seen, or perhaps subset the data to include a smaller set of genes that can be more easily seen.
Figure 6.a has been enlarged, and genes that are referred to in the text are singled out with increased text size. Furthermore the full list of differentially expressed genes has been included as a new Supplementary table 3.

Reviewer #3 (Remarks to the Author):

Holm et al. present a manuscript on "Neoantigen CD8 T cell recognition is broadened and activated in relation to disease control in the peripheral blood of patients with metastatic urothelial carcinoma following PD-L1 blockade". This study utilize a high-throughput screening approach to serially analyze CD8 T cell recognition of patient-specific neopeptides predicted from the pre-treatment tumor mutagenome in the peripheral blood of 24 patients with metastatic urothelial carcinoma treated with anti-PD-L1-therapy. The authors describe a rapid increase in the number of NART responses from pre-treatment to 3 weeks post-treatment initiation. The authors claim, that the overall neoepitope recognition breadth, not the estimated size of such CD8 T cell populations, was associated with clinical radiographic response. mRNA expression patterns of the tumor microenvironment pre-treatment were associated with increased NART responses. This is a fascinating approach, that can contribute significant knowledge to the field. However, the interpretation of the data raises several questions.

Major comments:

- 1) Abstract: „significant increase in the number of neoantigen-reactive CD8+ T cell (NART) responses“ remains unclear what is meant ... broadening of epitope specificity or increase in

numbers or increased functionality.

The abstract has been rephrased according to reviewers suggestion

- 2) Figure 2b: Why is the NART response so inconsistent over time? I assume the example in the figure was not the worst patient example. 4 out of 6 HLA-A2 restricted responses are detected only at one time point (not even all at one time), 2 at two out of 5 time points. Since T-cell responses are expected to be part of long-term immune responses, this raises questions about the validity of single NART events.

In the revised manuscript we have validated a large fraction of the observed T cell recognition using conventional fluorescent labeled MHC tetramers. Based on sample availability we could re-examine 65 NART responses. The data is included in Fig 2c and Suppl fig 3. Using this approach we could validate 50 of the responses with high confidence, and additional 9 responses were borderline to the detection limit.

Related to the consistency of the T cell responses over time, we have included a new Supplementary Table 1 listing the responses per patient, and divided over the different time-points. 45 out 148 NART responses were detected at multiple time points, while the remaining only at a single time-point. Many responses are of very low frequency, and hence minor changes in the lymphocyte compartment could result in the lack of detection of such low frequent responses.

- 3) Flowcytometry based validation of these responses with a detectable population using a single peptide MHC multimer would validate responses.

As mentioned above, such validation has been conducted, and is included in the new Supplementary Figure 3.

- 4) Page 5, line 96: "multiple patients" please specify.

This has been specified in the text - 18 of 22 patients.

- 5) Page 6, line 114 and Figure 3: Differences are given at time "3 weeks". Are there differences in NART at other time points?

No difference in number of NART responses were observed in any other timepoints than 3 weeks post-treatment. This has been rephrased.

- 6) What is the time point of response evaluation? Is it always the same time as the NART are detected? Is there a correlation when response has taken place and the NART detection?

As stated under comments for reviewer #1, further discussion with relevant references related to kinetics and clinical responses is now included in the revised manuscript, in the discussion.

Patients underwent tumor assessment at baseline and thereafter every 9 weeks for the first 12 months following cycle 1, day 1. Scans were only performed earlier if patients were clinically deteriorating to confirm progression. Interestingly, the median time to response in this trial (IMvigor210) was 2.1 months (95% CI 2.0-2.2), i.e. close to the time of first scan. Potentially there may have been tumor reduction prior to first scan, but this would not have been captured before first scan. We nevertheless believe that our observations are consistent with the concept of early

proliferative burst leading to early radiographic responses. This is further addressed in the revised discussion.

- 7) Can the authors provide evidence, that a NART detection at 3 weeks is responsible for RECIST criteria at 6 months?

We observe an association between the broadening of NART responses at 3 weeks and the treatment response. A causal effect cannot be determined, and the findings should be validation in other clinical cohorts as well. We have rephrased to clarify this where needed.

- 8) Flow data are not convincing. Parameters of T cell subpopulations Tem, Tcm, Tscm, Teff would be interesting rather than ki67. Exhaustion markers would be helpful to interpret the data.

We have revised the phenotype section for clarity. We have included bulk CD8, NART and VART distribution in Naïve/Tcm/Tem/TEMRA compartments in Supplementary Figure 6.g. The larger fraction of post-treatment NARTs are seen as Tem's, although insignificant between patient groups.

Additionally, looking at exhaustion markers, in addition to the single-parameter frequencies, we also quantified triple-positive CD57-CD45RA-GzmB bulk CD8 T cells, NARTs and VARTs between timepoints and patient groups in Supplementary Figures 6. e+f and observed no significant difference in either comparisons.

In this cohort the results suggest that NART expression of Ki67 and in part PD1 post-treatment are the key parameters associated to clinical outcome

Minors comments:

- 9) The manuscript is cumbersome to read. Abbreviations are not consistently used and/or explained. Text in the figures is often too small read it in a print out page.

We extensively revised the manuscript to ease reading and understanding for the reader and substantially revised figures label texts to improve quality and readability of figures and tables.

REVIEWER COMMENTS

Reviewer #1 (Remarks to the Author):

the authors have addressed my comments

Reviewer #2 (Remarks to the Author):

Summary: Holm et al. investigated the expansion of neoantigen specific CD8 T cell responses following PD-L1 blockade in blood of patients with metastatic urothelial carcinoma. Using their previously published technique for large-scale detection of antigen-specific T cells using peptide-MHCI multimers with labeled DNA barcodes, they were able to screen peripheral blood samples pre- and post PD-L1 blockade for neoantigen-reactive CD8 T cells (NARTs). The authors showed that 3 weeks after PD-L1 blockade there was an increase in the breadth, but not magnitude, of neoantigen specific CD8 T cell responses in complete responders but not in non-responders. In terms of T cell quality, responders had increased expression of PD-1 and Ki-67 compared to non-responders. Overall, the correlative observations may be worth exploring further to determine the dynamic changes of NARTs in response to PD-L1 blockade.

Major comments:

(1) The authors addressed our previous comments well and have either edited or added figures to their manuscript to answer questions posed previously.

(2) The first claim in the discussion may be over-reaching. 'First, we observed a rapid increase in the number of NART responses from pre-treatment to 3 weeks post-treatment initiation.' There is no figure that supports this claim with statistical significance. In figure 2, the authors compare within the same time point (either pre-treatment or 3 weeks post) but never compared each group (PD, SD, PR, CR) with itself between pre-treatment and 3 weeks post. If they compared within the same group, or total number of NART responses pre-treatment and 3 weeks post, and this was found to be significant then this claim would be well-founded.

(3) Line 310 'Third, expression of CD39 distinguished NARTs from bystander bulk CD8+ T cells and VARTs in the blood.' Evidence for this claim is not very convincing, in particular for use of the word 'distinguished'. There are no statistically significant differences between the bulk CD8s and NARTs as far as CD39 expression goes in figure 4h/i. In addition, the statistical difference in CD39 expression between the NARTs and VARTs looks like it might be driven by the few samples that have particularly high expression. It would be useful to decrease the range of the y-axis from 50 to 25 so that the spread of the data can be more easily assessed and the differences between groups better visualized. This claim is also repeated in lines 375-377, and needs to be revised.

(4) The heatmap in figure 6 is still difficult to interpret. The authors should consider reducing the genes displayed to gene sets of interest or key individual genes so that the expression of those genes can be clearly seen by the reader. There is no reason to include all other genes if names will not be provided. The full dataset can always be provided to researchers upon request.

Minor comments:

Reviewer #3 (Remarks to the Author):

The authors addressed the comments raised in the review

Response to REVIEWER COMMENTS, Revision 2

Reviewer #1 (Remarks to the Author):

The authors have addressed my comments.

Reviewer #2 (Remarks to the Author):

Summary: Holm et al. investigated the expansion of neoantigen specific CD8 T cell responses following PD-L1 blockade in blood of patients with metastatic urothelial carcinoma. Using their previously published technique for large-scale detection of antigen-specific T cells using peptide-MHCI multimers with labeled DNA barcodes, they were able to screen peripheral blood samples pre- and post PD-L1 blockade for neoantigen-reactive CD8 T cells (NARTs). The authors showed that 3 weeks after PD-L1 blockade there was an increase in the breadth, but not magnitude, of neoantigen specific CD8 T cell responses in complete responders but not in non-responders. In terms of T cell quality, responders had increased expression of PD-1 and Ki-67 compared to non-responders. Overall, the correlative observations may be worth exploring further to determine the dynamic changes of NARTs in response to PD-L1 blockade.

Major comments:

- 1) The authors addressed our previous comments well and have either edited or added figures to their manuscript to answer questions posed previously.
We thank the reviewer for the thoughtful comments.
- 2) The first claim in the discussion may be over-reaching. 'First, we observed a rapid increase in the number of NART responses from pre-treatment to 3 weeks post-treatment initiation.' There is no figure that supports this claim with statistical significance. In figure 2, the authors compare within the same time point (either pre-treatment or 3 weeks post) but never compared each group (PD, SD, PR, CR) with itself between pre-treatment and 3 weeks post. If they compared within the same group, or total number of NART responses pre-treatment and 3 weeks post, and this was found to be significant then this claim would be well-founded.

The reviewer makes an excellent point. In the results section, we state, "The change (delta) in number of detected NART responses between pre-treatment and 3 weeks post-treatment was calculated to better approximate patient-specific NART dynamics (Figure 3.d-e). A significant increase in Δ NART responses was observed for patients with disease control compared to those with PD (Figure 3.d, $p = 0.012$), with CR patients experiencing the largest Δ NART responses compared to PD patients (Figure 3.e, $p = 0.022$)." Thus, we have modified the statement in question to now say, "we observed an increase in NART responses from pre-treatment to 3 weeks post-treatment in patients with disease control."

- 3) Line 310 'Third, expression of CD39 distinguished NARTs from bystander bulk CD8+ T cells and VARTs in the blood.' Evidence for this claim is not very convincing, in particular for use of the word 'distinguished'. There are no statistically significant differences between the bulk CD8s and NARTs as far as CD39 expression goes in figure 4h/i. In addition, the statistical difference in CD39

expression between the NARTs and VARTs looks like it might be driven by the few samples that have particularly high expression. It would be useful to decrease the range of the y-axis from 50 to 25 so that the spread of the data can be more easily assessed and the differences between groups better visualized. This claim is also repeated in lines 375-377, and needs to be revised.

We thank the reviewer for this feedback. The sentences have been revised. Although the reviewer makes a good point, to facilitate the same scale of plotting between pre- and 3wk post-treatment figures, the Y-scale is kept at 50%.

- 4) The heatmap in figure 6 is still difficult to interpret. The authors should consider reducing the genes displayed to gene sets of interest or key individual genes so that the expression of those genes can be clearly seen by the reader. There is no reason to include all other genes if names will not be provided. The full dataset can always be provided to researchers upon request.

The heatmap in Figure 6.a is included to illustrate the full set of differentially expressed genes, which leads to a clustering of patients as given in the two colored rows above the heatmap. The selected, highlighted genes themselves would not provide the same clustering of patients, and consequently we find that the heatmap should include all the differentially expressed genes. The individual values can be found in the supplementary spreadsheet providing all values.

Reviewer #3 (Remarks to the Author):

The authors have addressed my comments.